# *Col4a3^-/-^* Mice on Balb/C Background Have Less Severe Cardiorespiratory Phenotype and SGLT2 Over-Expression Compared to 129x1/SvJ and C57Bl/6 Backgrounds

**DOI:** 10.3390/ijms23126674

**Published:** 2022-06-15

**Authors:** Camila I. Irion, Monique Williams, Jose Condor Capcha, Trevor Eisenberg, Guerline Lambert, Lauro M. Takeuchi, Grace Seo, Keyvan Yousefi, Rosemeire Kanashiro-Takeuchi, Keith A. Webster, Karen C. Young, Joshua M. Hare, Lina A. Shehadeh

**Affiliations:** 1Department of Medicine, Division of Cardiology, Leonard M. Miller School of Medicine, University of Miami, Miami, FL 33136, USA; cii4@med.miami.edu (C.I.I.); mmw154@miami.edu (M.W.); jmcondor@med.miami.edu (J.C.C.); tie5@med.miami.edu (T.E.); glambert@med.miami.edu (G.L.); jhare@med.miami.edu (J.M.H.); 2Leonard M. Miller School of Medicine, Interdisciplinary Stem Cell Institute, University of Miami, Miami, FL 33136, USA; ltakeuchi@med.miami.edu (L.M.T.); k1@miami.edu (K.Y.); rkanashiro@med.miami.edu (R.K.-T.); 3Department of Medical Education, Leonard M. Miller School of Medicine, University of Miami, Miami, FL 33136, USA; g.seo@med.miami.edu; 4Department of Molecular and Cellular Pharmacology, Leonard M. Miller School of Medicine, University of Miami, Miami, FL 33136, USA; 5Department of Ophthalmology, Cullen Eye Institute, Baylor College of Medicine, Houston, TX 77030, USA; kwebster@med.miami.edu; 6Department of Ophthalmology, Vascular Biology Institute, Leonard M. Miller School of Medicine, University of Miami, Miami, FL 33136, USA; 7Department of Pediatrics, Leonard M. Miller School of Medicine, University of Miami, Miami, FL 33136, USA; kyoung3@med.miami.edu

**Keywords:** Alport syndrome, SGLT2, cardiorespiratory phenotype, echocardiography

## Abstract

Alport syndrome (AS) is a hereditary renal disorder with no etiological therapy. In the preclinical *Col4a3^-/-^* model of AS, disease progression and severity vary depending on mouse strain. The sodium-glucose cotransporter 2 (SGLT2) is emerging as an attractive therapeutic target in cardiac/renal pathologies, but its application to AS remains untested. This study investigates cardiorespiratory function and SGLT2 renal expression in *Col4a3^-/-^* mice from three different genetic backgrounds, 129x1/SvJ, C57Bl/6 and Balb/C. male *Col4a3^-/-^* 129x1/SvJ mice displayed alterations consistent with heart failure with preserved ejection fraction (HFpEF). Female, but not male, C57Bl/6 and Balb/C *Col4a3^-/-^* mice exhibited mild changes in systolic and diastolic function of the heart by echocardiography. Male C57Bl/6 *Col4a3^-/-^* mice presented systolic dysfunction by invasive hemodynamic analysis. All strains except Balb/C males demonstrated alterations in respiratory function. SGLT2 expression was significantly increased in AS compared to WT mice from all strains. However, cardiorespiratory abnormalities and SGLT2 over-expression were significantly less in AS Balb/C mice compared to the other two strains. Systolic blood pressure was significantly elevated only in mutant 129x1/SvJ mice. The results provide further evidence for strain-dependent cardiorespiratory and hypertensive phenotype variations in mouse AS models, corroborated by renal SGLT2 expression, and support ongoing initiatives to develop SGLT2 inhibitors for the treatment of AS.

## 1. Introduction

Alport Syndrome (AS) is a progressive hereditary renal disease caused by mutations in type IV collagen genes. Around 80% of cases are associated with mutations in the X-linked *Col4a5* gene and other cases are associated with mutations in either the *Col4a3* or *Col4a4* genes [1]. It has an incidence of approximately 1 in 50,000 people that suffer renal dysfunction, sensorineural hearing loss and ocular abnormalities caused by progressive deterioration of the respective basement membrane (BM) functions [2,3,4]. Mouse models of autosomal recessive AS with mutations of *Col4a3* or both *Col4a3* and *Col4a4* genes exhibit glomerular basement membrane (GBM) lesions that mimic the human disease and present with delayed onset glomerulonephritis that progresses to end-stage renal failure (ESRF) and early death. During late stages of the disease, the mice model presented with glomerular pathology reminiscent of crescentic glomerulonephritis, together with a robust tubulointerstitial component, a characteristic also visible in human progressive chronic nephritis [5]. As with the human condition, in studies in animal models, including mice and dogs, the timing of disease onset and progression to ESRF that harbors identical mutations can be highly variable, effects that have been attributed to differential environmental influences and/or variation in genetic background [5,6,7,8]. Because AS is primarily a disease of the BM, the pathology impacts ocular and auditory functions as well as renal functions in both humans and murine models, again in a manner related to genetic background [9,10,11].

AS in both patients [12] and mouse models [10,13] involves progressively decreased GBM functions and glomerular filtration rate (GFR) that lead sequentially to hematuria, proteinuria and increased blood urea nitrogen (BUN) and creatinine, indicators of glomerular disease and BM dysfunction caused by accumulation of extracellular matrix, thinning, stretching, interstitial fibrosis and leakage that ultimately result in ESRF. The effects are attributed to compromised integrity of the glomerular capillary tufts caused by disrupted collagen networks that are increasingly sensitive to biomechanical stress. Loss of collagens α3/α5(IV) and increased levels of collagens αl/α2(IV), fibronectin and collagen VI are responsible for the structural and functional changes of the GBM in both mice and humans. Crosslinking of α1/α2(IV) collagen networks is much less than that of α3/α4/α5(IV) [14]. In common with multiple kidney diseases, the rate of progression of such glomerular disease and BM injury is augmented by hypertension in patients as well as AS mice [15,16,17,18]. Angiotensinogen, Ang-II and renin increase during AS progression in mice and humans, and ACE-inhibitors such as ramipril that block the renin angiotensinogen aldosterone system (RAAS) slow AS progression in mice and are the standard care for AS patients, beneficially slowing both renal and cardiovascular outcomes [16,17,19]. There have been several reports of aortic abnormalities in young males with AS, including aortic dilation, thoracic and abdominal aortic aneurysms and aortic dissection [20,21,22,23]. There is a potential molecular link between collagen IV α5 variants and aortic disease, since the collagen IV α5α5α6 network is normally expressed in the basement membranes of the aortic musculature in non-Alport subjects [24,25] and in wild type but not Alport mice [20]. If aortic disease is truly a complication of AS, very little is known about its incidence, prevalence or natural history (reviewed in [26]). Cardiovascular disease is a corollary and a frequent cause of death secondary to chronic kidney disease (CKD) wherein hypertension and hyperglycemia are driving forces and therapeutic targets especially on diabetic backgrounds (reviewed in [27]). Our group recently reported that *Col4a3^-/-^*-129J mice reproduce multiple features of heart failure with preserved ejection fraction (HFpEF), including symptoms of systemic hypertension and pulmonary congestion. Disease progression in this model, including cardiac performance and renal functions, was partially ameliorated by treatment with carvedilol, an α/β-adrenergic blocker, effects that were enhanced by exercise training [28,29]. A major goal of the present work is to create a comprehensive reference base for cardiorespiratory phenotypes associated with CKD within the context of murine AS.

Sodium-glucose cotransporter 2 (SGLT2) inhibitors represent a newer category of anti-hyperglycemic drugs that confer both nephroprotection and cardioprotection in patients with CKD and congestive heart failure, and are approved for the treatment of type 2 diabetes mellitus and its sequelae [27,30,31,32,33,34]. SGLT2 is the main glucose transporter situated in the nephron proximal tubules and its inhibitors lower blood glucose by blocking glucose reabsorption in the first convoluted tubule, thereby increasing excretion in the urine [35,36]. Clinical and animal studies indicate nephro- and cardio-protective properties of SGLT2 inhibitors related to blood pressure control, inflammation, glomerular epithelial to mesenchymal transition and ROS, which are independent of glycemic control [27,30,37,38,39]. The broad spectrum actions implicate crosstalk between SGLT2 inhibitors and the sympathetic nervous system, as well as with the RAAS [27,40]. Because of these properties, SGLT2 inhibitors are implicated, possibly in combination with ACE inhibitors, as potentially beneficial for AS [27,41]. To our knowledge, SGLT2 gene expression in AS of humans or animal models has not been reported; however, increased expression has been reported in tubular epithelial cells from urine of diabetic patients [42], although conflicting results have been reported in tissue samples [43,44]. Tubular epithelia of type 1 and type 2 diabetic mice had elevated SGLT2 levels [45], and mice fed high-fat, high-salt (HF/HS) diets were also reported to increase kidney SGLT2 gene expression, possibly in an Ang-II-dependent manner [46,47,48].

To investigate influences of genetic background on AS progression as it relates to cardiovascular disease, we quantified cardiac and cardiorespiratory functions, blood pressure, renal tubule function and kidney SGLT2 expression in male and female *Col4a3^-/-^* mice on three different backgrounds, including 129J, BALB/C and C57Bl/6. Previous studies comparing *Col4a3^-/-^* 129J with C57Bl/6 strain showed that genetic background can influence renal disease progression, consistent with variations seen in human AS pedigrees [5,27,49]. The results document, for the first time, a comprehensive context-dependent characterization of cardiorespiratory functions that parallel hypertension, renal tubule dysfunction and increased expression of SGLT2 in AS mice of both sexes. Results are discussed in terms of relations between genetic background and disease severity.

## 2. Results

Cardiorespiratory physiology was evaluated with the use of echocardiography, tail cuff for blood pressure, PV loops and WBP. Animals were analyzed separately by sex.

### 2.1. Different Genetic Background Affects Lifespan in Alport Mice

A Kaplan–Meier curve (Figure 1A) showed that *Col4a3^-/-^* 129x1/SvJ mice died prematurely (around 12 weeks) compared with Balb/C mice that survived up to 18 weeks old (*p* = 0.0029). It is known that C57Bl/6 mice have a longer lifespan (~30 weeks) compared to *Col4a3^-/-^* 129x1/SvJ mice [5,49] as replicated in our data (Figure 1A).

### 2.2. Echocardiography and Invasive Hemodynamics Revealed That Col4a3^-/-^ 129x1/SvJ Mice Displayed More Impaired Systolic Function and More Diastolic Dysfunction When Compared to Balb/C and C57Bl/6

#### 2.2.1. Systolic Function

Compared with their respective age-matched WT controls, both male and female 8-week-old *Col4a3^-/-^* 129x1/SvJ mice demonstrated mildly impaired systolic function. Males displayed decreased stroke volume (from 38.4 ± 4.97 to 30.06 ± 6.74 µL, *p* = 0.02) and alterations in myocardial deformation as evidenced by global longitudinal strain (GLS) analysis as shown in Figure 1, Table 1 (from −18.12% ± 4.04 to −12.85% ± 3.6, *p* = 0.02), with no changes in EF% (56.74 ± 10.08 for WTs and 54.04 ± 15.97 for Alport, *p* < 0.05). Female *Col4a3^-/-^* 129x1/SvJ mice displayed significant alterations in ejection fraction (EF) (from 64.75% ± 9.3 to 52.17% ± 10.73, *p* = 0.009), fractional shortening (FS%) (from 35.07 ± 6.65 to 26.56 ± 7.01, *p* = 0.03), increased end-systolic volume (ESV) (from 18.87 ± 5.59 to 27.42 ± 9.4 µL, *p* = 0.04) and alterations in myocardial deformation represented by GLS (*p* = 0.009) and global circumferential strain (GCS) (*p* = 0.001) as shown in Figure 1 and Table 2. Interestingly, only *Col4a3^-/-^* females from C57Bl/6 strain displayed a significant decrease in some parameters of systolic function, including EF% (from 58.48 ± 8.31 to 42.21 ± 8.49%, *p* = 0.0009), FS% (30.61 ± 6.17 to 20.64% ± 4.940, *p* = 0.004), increase in ESV (23.85 ± 7.12 to 42.53 ± 8.82 µL, *p* < 0.0001) and alterations in GCS (−21.32 ± 3.28 to −14.06% ± 4.42, *p* = 0.0016) (Figure 1, Table 2). Only *Col4a3^-/-^* Balb/C strain females exhibited decreased SV (*p* = 0.013) when compared to WT-Balb/C. Males did not show changes when compared to the WT control. The striking results observed in systolic parameters also showed that, independent of disease condition (WT/Alport), animals from different backgrounds can have various cardiac phenotypes. Changes in some parameters such as EF% (in females), SV (males) and CO (males), EDV (females) and GCS (male and female) were smaller in BalB/C when compared with C57Bl/6 (*p* < 0.05) and also 129J (*p* < 0.05) (Figure 1, Table 1 and Table 2). Remarkably, despite the reduction in EF% for females *Col4a3^-/-^* 129x1/SvJ (from 64.75 to 52.17) and *Col4a3^-/-^* C57Bl/6 (from 57.33 to 42.18%), values remained >40%, which represents an intermediate group for HF with preserved EF (HFpEF) [50].

#### 2.2.2. Diastolic Function

When we studied parameters related to diastolic function using transmitral inflow Doppler from an apical four-chamber view, males from *Col4a3^-/-^* 129x1/SvJ had impairment in left ventricular (LV) relaxation evidenced by prolongation of the isovolumetric relaxation time (IVRT) compared with WT controls (16.25 ms to 26.22 ms, *p* < 0.0001) and significant decrease in MV E velocity (665.76mm/s in WT vs. 480.88 mm/s in Alport mice, *p* = 0.024), reflecting a decrease in early ventricular filling (Table 1). This resulted in a decreased E/A ratio when comparing only WT vs. Alport (by Mann–Whitney test *p* = 0.043) as shown in Figure 2E and representative Figure 2F from WT-129J and *Col4a3^-/-^* 129x1/SvJ where the symmetrical shape of the E/A wave is quite evident. The myocardial performance index (MPI) or Tei index, a measure of global dysfunction and a useful index for assessing systolic and diastolic function, was also significantly increased in these mice (0.63 in WT vs. 1.06 in Alport, *p* < 0.001). Increased MPI is related to worsening LV diastolic function [51]. In *Col4a3^-/-^* 129x1/SvJ, female alterations were mild with increased IVRT (from 15.32 to 22.27 ms, *p* = 0.001) and MPI (by *t*-test *p* = 0.018) when compared with the control group (Figure 2, Table 2).

On Balb/C background only, females presented with increased IVRT (16.01 vs. 24.92 ms *p* < 0.0001), IVCT (15.36 vs. 22.12 ms, *p* = 0.04) and MPI (0.53 vs. 0.81, *p* = 0.0105). Similar to 129 strain, Balc/C males displayed a significant decrease in E/A ratio when comparing only WT with Alport (by *t*-test, *p* = 0.0104, Figure 2G,H) but no differences in IVRT and MPI were observed (Figure 2, Table 1).

For the C57Bl/6 strain, males and females exhibited mild changes in diastolic function with increased IVRT (15.38 vs. 21.25 ms, *p* = 0.016 in males and 17.44 vs. 22.73 ms, *p* = 0.046 in females). Females also displayed a significant increase in MPI when comparing WT to Alport (*p* = 0.006, Figure 2D). E/E′ did not change significantly for any of the strains (Figure 2B). Males and females Alport 129J presented with more exacerbated alterations in contrast to Balb/C and C57Bl/6, including a significantly higher IVRT (*p* < 0.05) and IVCT in males (*p* < 0.05), lower MV E (in males and females, *p* < 0.05) and consequently a lower E/A ratio (*p* < 0.05) and higher MPI (in males, *p* < 0.05) (Figure 2, Table 1 and Table 2).

#### 2.2.3. Pulmonary Artery and Aorta Flow

Doppler data analyses of the pulmonary artery and ascending aorta from all strains and sexes are described in Appendix A. Alport-129 mice showed only decreased time velocity integral to pulmonary flow (PV VTI) (from 30.1 to 23.39 mm) compared to WT. No other significant changes were observed in Alport mice from the three strains in comparison to their corresponding WT control. The data showed that 129J, Balb/C and C57Bl/6 presented different phenotypes for these parameters (not dependent on WT/Alport condition). Male mice from Balb/C background presented with lower PV peak velocity, PV peak gradient and PV peak pressure when compared with mice from 129J (*p* < 0.05) and C57Bl/6 strains (*p* < 0.05). Moreover, male mice from Alport-129J background presented with a significantly smaller AET, larger aortic valve peak velocity and peak pressure compared to Balb/C (*p* < 0.05) and C57Bl6 (*p* < 0.05) male mice.

Female mice from the 129J strain (WT and Alport) displayed a higher PV peak velocity (*p* < 0.001), PV mean gradient (*p* < 0.05), PV peak gradient (*p* < 0.05), increased PV peak pressure (*p* < 0.05) and alterations in PAT/PET ratio (*p* < 0.05) when compared to Balb/C strain (WT and Alport mice). Moreover, females from Balb/C strain presented with a prolonged AET, smaller aorta peak velocity (*p* < 0.05) and peak pressure (*p* < 0.001) compared to 129J and C57bl/6 background.

#### 2.2.4. Wall Thickness

When we studied cardiac dimensions via echocardiography, only males from the 129J background showed increased LV mass (WT = 4.82 mg/g vs. 5.68 mg/g for Alport mice, *p* < 0.001) when compared to WT-129J group. Left ventricle anterior wall (LVAW), left ventricle posterior wall (LVPW) and left ventricle internal diameter (LVID) did not change in these mice. Gravimetric findings obtained 2–5 days after echocardiography (Appendix A) confirm the increase in heart weight (HW) (WT = 5.31 mg/g vs. 6.6 mg/g for Alport mice, *p* < 0.001) for the 129J strain. These mice also presented with a decrease in body weight (BW) at 8 weeks old (21.81 to 17.62 g, *p* < 0.0001), similar to males from the Balb/C background (23.25 to 19.89 g, *p* < 0.05). Alport C57Bl/6 male mice did not show any significant differences compared to WT-C57Bl/6. Regarding the females, mice from 129J displayed a significant increase in LV area in systole (10.36 to 13.04 mm^2^, *p* = 0.0135) as shown in Table 2. This is associated with an increased HW corrected by BW (*p* < 0.05, Appendix A). In contrast, females from Alport Balb/C strain did not show any differences. Female Alport C57Bl6 mice displayed increased left ventricular area in systole (From 12.14 to 15.8 mm^2^, *p* = 0.005) combined with an increased LVID in systole (*p* = 0.0001) and in diastole (*p* = 0.0043).

#### 2.2.5. Pressure-Volume (PV) Loop

Data obtained from invasive hemodynamic measurements allow for assessment of cardiac function and confirmed alterations observed by echocardiography. Two-month old male Alport 129J background mice had a significantly increased end-diastolic pressure-volume relationship (EDPVR, *p* = 0.0329) and increased arterial elastance (Ea, *p* < 0.0001), suggesting impaired LV relaxation in *Col4a3^-/-^* mice (Figure 3A,B, Appendix A). dP/dT_max_, an index of the global LV contractile state during isovolumic contraction, was also increased (*p* = 0.027 by *t*-test) along with a significant decrease in SV (*p* = 0.024), EF (*p* = 0.01) and consequent decrease in stroke work (SW) (*p* = 0.023) (Figure 3D). Males from BalB/c and C57Bl/6 background did not present evident changes related to diastolic dysfunction. Male mice from Balb/C background only displayed increased end-systolic pressure-volume relationship (ESPVR, *p* = 0.036 by *t*-test) and *Col4a3^-/-^* mice from C57Bl/6 had increased ESPVR (*p* = 0.007), decreased SW (*p* = 0.012), CO (*p* = 0.0088) and end-systolic pressure (*p* = 0.029). Invasive LV catheterization in *Col4a3^-/-^* 129J female mice, as represented in Figure 3, revealed increased EDPVR (*p* = 0.027 by *t*-test), increased end-diastolic pressure (*p* = 0.0027 by *t*-test) and markedly prolonged time constant of LV relaxation (Tau; *p* = 0.0083 by *t*-test) (Appendix A), supporting diastolic dysfunction. Furthermore, Alport C57Bl/6 showed a significant increase in EDPVR (*p* = 0.0086) associated with decreased SW (*p* = 0.0016), CO (*p* = 0.0185), SV (*p* = 0.0086), end-systolic pressure (*p* = 0.0135) and EF (*p* = 0.031). Alport Balb/C did not show significant changes in cardiac function when compared with WT control. Consistent with observations in echocardiography, hemodynamic data showed variations in cardiac function between each strain. This suggests that mouse background influences cardiac function. Complete invasive hemodynamic measurements are described in Appendix A.

### 2.3. Only Col4a3^-/-^ 129x1/SvJ Mice Displayed Alteration in Blood Pressure by Tail Cuff

As shown in Figure 4A, male *Col4a3^-/-^* 129x1/SvJ mice presented with an elevated systolic blood pressure (SBP) compared to WT 129x1/SvJ (from 110.41 mmHg ± 12.57 to 133.13 mmHg ± 16.71, *p* = 0.014) and compared to Balb/C strain (*p* = 0.032) and C57bl/6 (*p* = 0.048) strains. Female *Col4a3*^-/-^129x1/SvJ had elevated systolic blood pressure (SBP) (110.55 mmHg ± 9.50 to 133.67 mmHg ± 18.47, *p* = 0.012), diastolic blood pressure (DBP) (from 52 mmHg ± 10.98 mmHg, *p* = 0.023) and mean pressure (from 72.11 ± 8.921 mmHg to 93 ± 18.92 mmHg, *p* = 0.044). Elevated blood pressure is frequently found in Alport syndrome patients [52]. No changes in blood pressure were detected in mice from Balb/C and C57Bl/6 background.

### 2.4. Respiratory Capacity Is Changed in Col4a3^-/-^ on 129x1/SvJ, Col4a3^-/-^ C57Bl/6 and Balb/C Backgrounds

#### 2.4.1. Whole Body Plethysmography (WBP)

Using WBP, our data demonstrated that Alport syndrome significantly impacted parameters of respiratory function (Figure 5 and Figure 6). Male *Col4a3^-/-^* 129x1/SvJ mice showed decreased respiratory frequency (RF) (from 275.9 bpm ± 56.9 to 226.8 bpm ± 49.2, *p* = 0.045 by *t*-test), increased tidal volume corrected for body weight (TV/BW) (from 0.0094 ± 0.00108 to 0.011 ± 0.0012 mL/g, *p* = 0.001 by *t*-test), prolonged expiratory time (Te) (from 0.17s ± 0.04 to 0.22s ± 0.04, *p* = 0.011 by *t*-test), and end expiratory pause (EEP), the pause between the end of one breath (end expiration) and the beginning of the next breath (from 16.34 ± 5.91 to 34.89 ± 34.89 ms, *p* = 0.006 by *t*-test) and increased Rpef ratio (from 0.24 ± 0.04 to 0.30 ± 0.07, *p* = 0.0289 by *t*-test). Rpef is the ratio of time to peak expiratory flow (PEFb) relative to expiratory time (Te). Increased Rpef ratio has been reported in hypoxic conditions [53]. These mice also had prolonged relaxation time (Tr), time required to exhale 65% of breath volume (from 0.089 s ± 0.022 to 0.124 s ± 0.028, *p* = 0.0036), decreased end inspiratory pause (EIP) (from 6.65 ± 0.18 to 2.45 ms ± 0.26, *p* = 0.05 by *t*-test), time of pause (TP) (from 15.51 ± 2.68 to 11.74 ± 2.57%, *p* = 0.012), enhanced pause (Penh), an indirect measure of airway resistance (ref) (from 0.79 ± 0.26 to 0.59 ± 0.14, *p* = 0.0408, by *t*-test) and decreased EF50% (from 0.15 ± 0.04 to 0.11 mL/s ± 0.02, *p* = 0.012 by *t*-test). EF50% represents the flow rate at which 50% of the tidal volume has been expelled in an individual breath. Males from C57Bl6 displayed alterations similar to 129J mice, including the significantly prolonged end expiratory pause (EEP) (from 46.74 ± 20.95 to 67.27 ± 17.84 ms, *p* = 0.04), increased Rpef ratio (from 0.18 ± 0.04 to 0.24 ± 0.05, *p* = 0.0047 by *t*-test), prolonged relaxation time (Tr), (from 0.08 ± 0.02 to 0.10 ± 0.02, *p* = 0.014 by *t*-test), decreased EIP (from 2.33 ± 0.15 to 2.19 ms ± 0.16, *p* = 0.05 by *t*-test), time to pause (from 17.33 ± 1.93 to 14. ± 2.17%, *p* = 0.0065, by Mann–Whitney), enhanced pause (Penh) (from 1.09 ± 0.31 to 0.68 ± 0.15, *p* = 0.02) and decreased PAU (from 1.41 ± 0.31 to 1.1 ± 0.19 *p* = 0.0167). This last parameter provides a time comparison of early and late expiration, an indicator of bronchoconstriction [54]. In contrast to other strains, Alport males from Balb/C background did not show significant alterations in respiratory function (Figure 5).

The breathing patterns also changed for female mice when compared to WT controls of the same age. Female *Col4a3^-/-^* 129x1/SvJ mice displayed decreased respiratory frequency (RF) (from 279.36 bpm ± 45.73 to 232.47 bpm ± 37.9, *p* = 0.034 by *t*-test), decreased minute volume (MVb), the volume of gas inhaled or exhaled per minute (from 51.96 mL/min ± 7.83 to 44.35 mL/min ± 8.11, *p* = 0.047, by *t*-test) and PEFb (from 2.40 mL/s ± 0.32 to 1.85 mL/s ± 0.3, *p* = 0.0015 by *t*-test). Expiratory time was significantly prolonged in these mice (from 0.15 ± 0.028 to 0.2 ± 0.042, *p* = 0.0138 by *t*-test), along with increased relaxation time (from 0.078s ± 0.016 to 0.107s ± 0.021, *p* = 0.0085), decreased time of brake (TB) at the end of inspiratory phase (percentage of the breath occupied by transition from inspiration to expiration (from 5.97 ± 0.75 to 4.72 ± 1.38, *p* = 0.012) and decreased EF50% (from 0.159 ± 0.035 to 0.113 mL/s ± 0.023, *p* = 0.0057 by *t*-test). Alport female mice from C57Bl/6 background presented with increased TVb (from 0.24 mL ± 0.031 to 0.26 mL ± 0.024, *p* = 0.022 by *t*-test) and TVb/BW (from 0.010 mL/g ± 0.002 to 0.012 mL/g± 0.001, *p* = 0.0043 by *t*-test), increased Rpef ratio (from 0.19 ± 0.038 to 0.29 ± 0.095, *p* = 0.0021), PIFB (from 6.53 mL/s ± 1.37 to 7.89 mL/s ± 1.16, *p* = 0.025), prolonged relaxation time (Tr), (from 0.078s ± 0.014 to 0.101s ± 0.013, *p* = 0.0135 by *t*-test) and TB (from 5.63 ± 0.99 to 6.23 ± 1.04, *p* = 0.043 by *t*-test). However, TP (from 17.76 ± 2.35 to 14.78 ± 2.68, *p* = 0.0070 by *t*-test), Penh (from 0.94 ± 0.23 to 0.68 ± 0.14, *p* = 0.0041 by *t*-test) and PAU (from 1.33 ± 0.22 to 1.10 ± 0.19, *p* = 0.0165 by *t*-test) were decreased. In comparison to males, Alport Balb/C females exhibited alterations in respiratory function including decreased minute volume (MVb) (from 72.13 ± 13.87mL/min to 57.92 ± 14.84 mL/min, *p* = 0.015, by *t*-test), MVb/BW (from 4.0 ± 0.52 mL/min/g to 3.39 ± 0.79 mL/min/g, *p* = 0.033, by *t*-test) and decreased PEFb (from 3.61 mL/s ± 0.60 to 2.76 mL/s ± 0.53, *p* = 0.0024). RPef was increased (from 0.20 ± 0.042 to 0.27 ± 0.076, *p* = 0.0097 by *t*-test) in addition to increased TB (from 6.04 ± 0.58 to 6.57 ± 0.63, *p* = 0.030 by *t*-test) and decreased EF50% (from 0.205 ± 0.044 to 0.156 mL/s ± 0.04, *p* = 0.028). As expected, differences between strains were observed (Figure 5 and Figure 6). Also, rejection index (Rinx), the percentage of breaths rejected before a breath is accepted and an internal control of animal behavior or movement in the chamber, demonstrated that mice from 129J strain (both Alport and WT) were calmer compared to Balb/C (*p* < 0.05) and C57Bl/6 strains (*p* < 0.05). It was evident that Balb/C was more excited than others. This observation was supported by higher levels of rinx in the Balb/C mice that were more anxious compared to others (Appendix A), and previous report that suggested that Balb/C mice are more anxious and less social compared to C57Bl/6 [55].

#### 2.4.2. Diaphragm Ultrasonography and Gravimetry from the Lungs

Non-invasive ultrasonography of the diaphragm was used to evaluate diaphragm performance. No significant changes were observed for male and female mice from all strains (Appendix A). A trend was observed only for males 129J (decreased from 0.83 mm ± 0.10 to 0.57 mm ± 0.06). Gravimetry data showed increased LW/BW only for male Alport-129J mice (from 6.11 mg/g ± 1.04 to 7.47 mg/g ± 0.99, *p* = 0.006) suggesting pulmonary congestion in those mice (Appendix A).

### 2.5. Blood Urea Nitrogen (BUN) and Creatinine (CRE) Were Increased in Alport Mice Compared to WT Control Mice

Increased levels of BUN and CRE have been reported in renal disease and are used for screening of renal function [56]. Plasma levels of BUN and CRE of all Alport mice groups (male and females) were higher when compared with WT controls. The levels of BUN and CRE in 8-week-old male *Col4a3^-/-^* 129x1/SvJ mice were 139.22 ± 107.92 mg/dL and 0.70 ± 0.49 mg/dL, respectively, compared to values in their respective WT control (35.56 mg/dL ± 11.46 and 0.28 ± 0.016, *p* < 0.05 by Mann–Whitney). Female mice had BUN levels of 21.8 mg/dL ± 1.09 for WT and 123 mg/dL ± 46.63 for Alport (*p* = 0.03) and CRE levels of 0.27 ± 0.083 for WT vs. 0.62 ± 0.31 for Alport mice (*p* = 0.07 by Mann–Whitney). For male Balb/C mice, plasma values were BUN of 22.63 mg/dL ± 3.96 for WT and 82.2 mg/dL ± 33.28 for Alport (*p* = 0.002) and CRE of 0.28 mg/dL ± 0.06 for WT and 0.56 mg/dL ± 0.23 for Alport (*p* = 0.0071 by *t*-test). Females from Balb/C background displayed higher levels of BUN (18.20 mg/dL ± 2.39 vs. 109 ± 63.98 mg/dL, *p* = 0.002) and CRE (0.25 mg/dL ± 0.049 vs. 0.47 mg/dL ± 0.25, *p* = 0.07 by Mann–Whitney). For male C57Bl/6 mice, differences were significant between Alport and WT for both BUN (from 26.6 mg/dL ± 3.53 to 94.71 mg/dL ± 52.09, *p* = 0.006) and CRE levels (from 0.29 mg/dL ± 0.005 to 0.61 mg/dL ± 0.39, *p* = 0.01). Female C57bl/6 BUN levels were significantly higher when compared to WT (from 28.87 mg/dL ± 5.22 to 117.42 mg/dL ± 82.38, *p* = 0.0003 by Mann–Whitney). Similarly, CRE values were also higher (0.28 mg/dL ± 0.0088 to 1.03 mg/dL ± 1.45, *p* = 0.002 by Mann–Whitney). BUN levels > 60 mg/dL, as shown by all Alport groups, are consistent with inefficient removal of urea by the kidney [57]. No changes where observed on glucose levels when comparing Alport groups with the corresponded WT group (Figure 7).

### 2.6. Genetic Background Affects SGLT2 Expression in the Kidney

SGLT2 expression was measured in kidneys from all strains by immunofluorescence. As shown in Figure 8 and Figure 9, histopathological analyses revealed both male and female Alport mice exhibited increased SGLT2 expression in the renal cortex compared to their respective WT control. Differences in SGLT2 expression in males from 129J strain were 1.28% ± 0.24 vs. 8.33% ± 1.8 (*p* < 0.0001), while Balb/C males displayed changes from 2.24% ± 0.56 to 5.46% ± 1.76 (*p* = 0.0129) and C57Bl/6 from 2.99% ± 1.04 to 10.33% ± 1.5 (*p* < 0.0001). Females from 129J strain showed a significant increase in the expression of SGLT2 from 1.27% ± 0.24 to 7.85% ± 2.93 (*p* = 0.0012). Balb/C strain revealed a minimal increase from 1.61% ± 0.39 to 3.7% ± 0.74 (*p* > 0.05) and C57Bl/6 from 2.02% ± 0.36 to 12.16% ± 3.64 (*p* < 0.0001). Interestingly, when compared to Alport mice from different strains, Balb/C mice had a significantly small expression of SGLT2 in the kidney when compared with male mice from 129J strain (*p* = 0.038) and C57Bl/6 strain (*p* = 0.0004). A similar trend was observed in females. The expression of SGLT2 for Balb/C mice was 3.7% ± 0.74, a significantly lower amount when compared to mice from 129J strain (7.85 ± 2.93, *p* = 0.0489) and C57Bl/6 strain (12.16 ± 3.64, *p* < 0.0001). Differences between Alport 129J female and Alport C57Bl/6 females were also observed (*p* = 0.0286). In data not shown, Sglt2 expression was not detected in the hearts.

## 3. Discussion

Influences of genetic background on AS progression have been reported in patients and animal models [5,49,58,59]. Specific contributions of immune system modulation and transcription factors involved in cell proliferation and survival, such as signal transducer and activator of transcription 3 (STAT3), have also been described [13,60]. Recently, our group reported a *Col4a3^-/-^* mouse model of HFpEF secondary to CKD that implicated osteopontin as an etiological effector and carvediol, an α/β adrenergic antagonist, as therapeutic [10,28]. Despite decades of investigation, an established cure for AS remains elusive, and standard treatment for patients is limited to ACE inhibitors that slow disease progression when applied at an appropriate early stage before onset of ESRF [16,17,19]. SGLT2 inhibitors have demonstrative nephroprotection and cardioprotection in patients with CKD, especially secondary to type 2 diabetes, and are promising candidates for AS [27,61]. Animal models have been indispensable in defining etiologies, cardiorenal communications and potential therapeutic targets of AS. Here, we investigated the differences in cardiorespiratory phenotypes of AS mice with a *Col4a3^-/-^* mutation on three different genetic backgrounds, and related changes of blood pressure, renal disease as reflected by increased plasma BUN/creatinine and SGLT2 expression in the kidneys. Our results confirm major roles for genetic background in determining cardiorespiratory phenotypes in this model and are consistent with contributions of hypertension and possibly SGLT2 as secondary drivers of the cardiorenal pathophysiology.

As expected, all three strains of mice displayed decreased glomerular function as indicated by increased plasma BUN and creatinine (Figure 7), defining characteristics of AS [13,62]. Whereas relatively minor differences could be attributed to genetic background or sex, a significant feature of BUN and creatinine expression was the large variation within each AS group relative to the tight grouping of wild types. Such variation of blood and urine kidney disease markers in these models is quite common [5,63], and reflects differences in individual subject responses to the same mutation, independent of genetic background and consistent with a highly interactive, multifactorial etiology. Interestingly, different laboratories have reported major difference in survival of mice with identical mutations and genetic backgrounds, again attesting to the sensitivity of the phenotypes to environmental conditions [63]. We did not observe major differences in blood glucose levels between AS groups and controls, consistent with an absence of diabetic nephropathy that is typically not a feature of early stage AS in mouse models or humans. Comprehensive cardiorespiratory analyses revealed more severe phenotypes and shortest survival times of *Col4a3^-/-^* mice on a 129x1/SvJ background, independent of sex relative to the other strains. These results are consistent with previous reports that found no sex difference of renal disease progression in *Col4a3^-/-^* mice on a 129/SvJ background [64]. Female *Col4a3^-/-^* mice on C57Bl/6 and Balb/C backgrounds trended towards more severe phenotypes compared with males, consistent with work that described sex differences [62,64,65]. Blood pressure analyses indicated significantly increased SBP of male and female *Col4a3^-/-^* mice only on a 129x1/SvJ background and significantly increased DBP only in *Col4a3^-/-^*-129x1/SvJ females (Figure 4). These results are consistent with the well-known role of hypertension in driving cardiorenal disease, and the success of ACE inhibitors to slow CDK and AS progression of patients as well as in mouse models (reviewed in [27,66,67]). Early activation of endothelin signaling and therapeutic benefit of ET_A_R blockade by sitaxentan, a selective inhibitor in AS mice, also support critical roles for hypertension in GBM injury and disease progression [68].

Whereas we have not attempted to investigate putative strain-specific AS modulating genes, early work comparing differences in disease progression of the *Col4a3^-/-^* mutation on 129X1/SvJ versus C57BL/6 backgrounds suggested linkage markers on chromosomes 9 and 16 [5]. Other studies implicated strong ectopic deposition of α5α6(IV) collagen in the GBM of *Col4a3^-/-^* mice on a C57BL/6J but not 129/Svj background, as potentially protecting the C57BL/6J GBM [69]. However, more recent studies that targeted both α3 and α6 collagen IV chain expression on 129Sv and C57BL/6J backgrounds reported no effect of enhanced expression of α5/α6 collagen on disease progression or renal pathology of the *Col4a3^-/-^* mutation in either strain, although C57BL/6J mice were still significantly protected [63]. Kang et al., [59] reported similar expression of α1/α2(IV) chains in *Col4a3^-/-^* mice on C57BL/6J or 129/Sv backgrounds, undetectable expression of α3/α4(IV) chains, and expression of α5/α6(IV) chains only in *Col4a3^-/-^*-C57BL/6J mice. Murata et al., [63] reported similar findings with respect to collagen gene expression, and further described expression of α5/α6 (IV) chains in the GBM of autosomal recessive patients with homozygous mutations in COL4A4, suggesting relevance of the condition to human AS. A rat Col4a5 deficiency model that conferred severe AS symptoms with rapid progression expressed no α3/α4(IV) or α5/α6(IV) but normal expression of α1/α2(IV) [62], similar to human AS where α1/α2(IV) is unaffected or increased by AS Col4 mutations [1]. Taken together, the results suggest that augmented deposition of alternate α-IV chains in *Col4a3^-/-^* mice does not contribute to the control of genetic background on AS severity. Because of the complex signaling pathways that determine integrity of the GBM, strain-specific modifier genes are likely to be diverse. For example, Falcone et al., identified differential, strain-specific expression of podocyte-specific genes and podocyte morphology as causal for AS progression and severity of the *Col4a3^-/-^* mutation in C57BL/6J versus C3H.Pde6b+ mice [58]. CD151-null mice with defective α3β1 integrin binding affinities but normal GBM type IV collagen composition, present with a phenotype that closely mimics that of *Col4a3^-/-^* mice with progressive renal dysfunction associated with GBM remodeling and functional deterioration [70]. The results suggest a diverse repertoire of potential strain/lineage-specific modifier genes that are not limited to expression of type IV collagens.

The present results support our previous work that implicated cardiorenal factors in the etiology of HFpEF including trends of slightly lower EF in all AS groups [28,29]. Similarly, Neuburg et al. [49] described systolic dysfunction in 10-week-old 129Sv *Col4a3^-/-^* mice and elevated systolic, diastolic and mean BP in both 129Sv and B6 mice. In our hands, only *Col4a3^-/-^* mice on 129J background were hypertensive and demonstrated increased LV mass. In contrast to AS on the 129J strain, C57Bl/6 mice displayed trends of systolic dysfunction, including decreased EF and FS, increased ESV and impaired GCS. PV loop analysis revealed that male *Col4a3^-/-^* mice from C57Bl/6 displayed increased ESPVR, decreased SW, CO and end-systolic pressure. Male and female C57Bl/6 mice exhibited more modest changes in diastolic function, reflected by increased IVRT and MPI. Females from the *Col4a3^-/-^* Balb/C strain showed the mildest cardiorespiratory phenotypes that were limited to decreased SV, and increased IVRT, IVCT and MPI.

Despite intensive study, to our knowledge respiratory dysfunctions of AS patients have not been documented, and there are no reports on the respiratory capacity of AS mice. Here we report significant alterations in inspiratory and expiratory parameters of both male and female *Col4a3^-/-^* 129x1/SvJ and C57Bl/6 mice. It has been reported that mice with significant respiratory compromise exhibit reduced respiratory frequency and tidal volume [71,72]. Interestingly, while the most pronounced blunting of respiratory drive was seen in male *Col4a3^-/-^* 129x1/SvJ mice, it was compensated by increased tidal volume. Male *Col4a3^-/-^* 129x1/SvJ mice also sustained the largest increase in Te, and reduced EF50%, suggesting airway narrowing. Additionally, male 129x1/SvJ mice displayed increased lung weight, suggesting pulmonary congestion, and significantly decreased time velocity integral to pulmonary flow (PV VTI). Diaphragm performance was slightly decreased in these mice when compared with controls. Unlike their male counterparts, female *Col4a3^-/-^* Balb/C mice exhibited alterations in respiratory function similar to their 129x1/SvJ and C57Bl/6 counterparts. While minute volumes of male *Col4a3^-/-^* 129x1/SvJ mice were maintained by increased tidal volume, females were unable to compensate, indicating more severe dysfunction. Previous studies have suggested that increases in tidal volume during diseased states pull the airways open and lower airway resistance by airway-parenchymal interdependence. In our study, female *Col4a3^-/-^* 129x1/SvJ mice showed severe airway constriction as evidenced by a significant increase in Te, along with marked reductions in peak expiratory flow and EF50%.

SGLT2 expression in the renal cortex was increased relative to controls in all AS groups irrespective of sex. AS on a Balb/C background were the outliers, displaying significantly lower expression of SGLT2 compared with the other two strains (Figure 8 and Figure 9). *Col4a3^-/-^* Balb/C mice also present with the least severe cardiorespiratory phenotype. Although we did not investigate mechanisms of SGLT2 regulation, it is noteworthy that SGLT2 gene expression is induced by diabetic environments in mice and humans, by high-fat/high-salt diets in mice, and by high-glucose culture of glomerular epithelial cells. Ang-II is implicated as a possible mediator in these studies [42,45,46,47,48]. Interestingly, whereas *Col4a3^-/-^* Balb/C mice presented the mildest cardiorespiratory phenotype and significantly lower SGLT2 expression, levels of plasma BUN and creatinine, indices of renal tubule injury, while highly variable in all *Col4a3^-/-^* mice, were not significantly affected by genetic background at the time points tested, suggesting a quantitative/temporal disconnect between renal and cardiorespiratory defects in these mice. Recent studies attempting to define the far-reaching properties of SGLT2 inhibitors on cardiorenal disease suggest bidirectional interactions between SGLT2 and the sympathetic nervous system (reviewed in [40]). Adrenergic pathways are also implicated in the beneficial effects of SGLT2 inhibitors for patients with chronic heart failure [73], and may be part of the mechanism of action of cardioprotection by carvedilolin *Col4a3^-/-^*-129x1/SvJ mice [28].

In conclusion, our results confirm major roles for genetic background on AS progression in mouse models as they relate to cardiovascular disease, cardiorespiratory functions, blood pressure and renal tubular SGLT2 expression in male and female *Col4a3^-/-^* mice. To our knowledge, except for a rare case of X-linked AS associated with diffuse leiomyomatosis [74], cardiorespiratory alterations associated with AS have not been described in patients or animal models. Our results document, for the first time, increased expression of SGLT2 in AS mice of both sexes, the levels of which coincided with genetic background and cardiorespiratory disease severity. AS mice on a 129x1/SvJ background presented more severe cardiorespiratory dysfunction, higher systolic blood pressure (and diastolic BP in females), and higher SGLT2 expression relative to the Balb/C background, and there was evidence of sex-related quantitative differences in some parameters. The results represent the first comprehensive analysis of cardiorespiratory phenotypes and the first evidence for regulation of SGLT2 expression associated with AS, and a possible influence of genetic background.

## 4. Materials and Methods

### 4.1. Animals

All procedures involving animals were approved by the Institutional Animal Care and Use Committee at the University of Miami, conforming to NIH guidelines (IACUC protocol 20-118). *Col4a3^-/-^* (Alport) mice on 129X1/SvJ background were purchased from Jackson Laboratory and crossed in-house for at least 10 generations with wild type (WT) mice from C57Bl/6 or BALB/c background to generate Alport mice from each strain. *Col4a3^-/-^* mice on 129X1/SvJ, C57Bl/6 and BALB/c background and their wild type littermates were used for all the physiological experiments. It is known that disease progression and lifespan in Alport mice from different genetic strains are not similar. For Alport mice on 129J and BALB/c background, studies were performed at 8 weeks of age. For Alport mice on the C57/Bl6 background, studies were performed at 20 weeks of age. Male and female mice were housed under specific pathogen-free conditions in temperature- and humidity-controlled rooms with a 12 h light-dark cycle. Mice were provided with water and food ad libitum. Dissections were performed under deep anesthesia (100 mg/kg ketamine and 20 mg/kg Xylazine via intraperitoneal injections) to collect hearts, lungs, kidneys and plasma.

### 4.2. Echocardiography, Strain Analysis and Diaphragm Ultrasonography

Morphology and cardiac function were assessed using Vevo2100 imaging system (VisualSonics, Toronto, ON, Canada) with a MS400 linear array transducer. Mice were shaved with depilatory cream one day prior to experiments. Briefly, mice were anesthetized with 2.5–3% isoflurane at 0.8 L/min flow rate and maintained with 1–1.5% isoflurane. Following anesthesia, the mice were fixed in a supine position on a pad with an integrated temperature sensor, heater and ECG electrodes. Both heart rate and body temperature were monitored constantly (maintained around 37 °C) during measurement. We used parasternal short (at the level of midpapillary muscles) and long-axis view to obtain two-dimensional B-mode and M-mode images. Cardiac function parameters examined in B mode include: area in systole, area in diastole, endocardial Major (d, s), epicardial major (d, s). In M-mode: included ejection fraction (%EF), stroke volume (SV), cardiac output (CO), % fractional shortening (%FS), left ventricle volume (d, diastole and s, systole). To study cardiac dimensions, we obtained (from M-mode view): interventricular septum (IVS) thickness (d, s), left ventricular anterior wall thickness (LVAWd and s), left ventricular posterior wall thickness (LVPWd and s), and left ventricular internal dimension (LVIDd and s), LV mass; LV mass corrected (LV mass/body weight). Diastolic function was assessed from the apical four-chamber view, by pulsed-wave Doppler analysis of mitral valve inflow and tissue Doppler imaging (TDI). Parameters included MV early wave peak (E), MV atrial wave peak (A), no flow time (NFT), aortic ejection time (AET), isovolumic relaxation time (IVRT), isovolumic contraction time (IVCT), E wave deceleration time (DT), myocardial performance index (Tei index), E′ wave (motion of the mitral annulus during early diastolic filling of the LV), A′ wave which originates from atrial systole during late filling of the left ventricle. Pulmonary valve (PV) peak velocity, PV peak pressure, aortic valve (AV) peak velocity and AV peak pressure were also measured. Strain analysis was performed from B-Mode for long and short axis views. The endocardial borders were traced in the end-systolic frame of the 2D images to obtain maximal opposing wall delay, global longitudinal strain (GLS, for long axis) and global circumferential strain (GCS, for short axis).

The amplitude of the diaphragm movement (in millimeters) was measured according to Whitehead et al. [75] using the same probe (M400) and M-mode image window. All data were obtained from three cardiac cycles and averaged using VevoLAb 3.3.3 software (Visual Sonics, Toronto, ON, Canada).

### 4.3. Tail-Cuff Blood Pressure Measurements

Blood pressure (BP) of conscious mice was recorded using a noninvasive tail-cuff method (BP-2000 Series II, Visitech Systems, Apex, NC, USA). The protocol was designed based on the manufacturer’s recommendations. Animals were placed in the restraining chambers on a warm platform and the occlusion cuff was placed at the base of the tail. The diastolic and systolic pressure was determined by monitoring the vessel dilation as the occlusion cuffs inflated (inflated to 250 mm Hg and deflated over 20 s). Prior to the experiment, mice were acclimatized for 4 consecutive days. On the following day (day 5), at least 10 to 20 measurements were collected and averaged after 5 preliminary cycles (not used in the analyses) in order to allow the animals to warm up. The systolic BP (SBP), diastolic BP (DBP), mean and pulse (BPM) were obtained. BP was performed in a proper environment (RT, lightning, and noise-free atmosphere) and at the same time each day.

### 4.4. Whole Body Plethysmography for Assessment of Lung Function

Respiratory function was monitored unrestrained in conscious mice by the Buxco small animal whole-body plethysmography system and FinePoint software (Data Science International, New Brighton, MN, USA). Prior to the measurements, each mouse was placed in separate cylindrical chambers for around 30 min/day for 3 consecutive days. On the final day (day 4), mice were acclimatized in the chamber for 10 min, then respiratory parameters were measured continuously for the next 10 min [76]. The system was calibrated prior to each recording session. Multiple parameters were recorded: respiratory frequency (breaths/min), tidal volume (TV), tidal volume (TV), minute volume (MV), enhanced pause (Penh), pause (PAU), inspiratory time (Ti), peak inspiratory flow (PIF), expiratory time (Te), peak expiratory flow (PEF), relaxation time (Tr), tidal midexpiratory flow (EF50) and total expiratory time (Rpef).

### 4.5. Hemodynamics and Pressure-Volume Loops

After completion of all echocardiography, pulmonary measurements and tail-cuff measurements, intra-arterial pressure-volume (PV) loops were performed on the mouse groups as described above [29]. Mice were anesthetized with isoflurane and placed on an operating surface maintained at 37 °C. A midline incision was made in the neck to allow for endotracheal intubation. The anesthesia was maintained by intubation with 2% isoflurane and mechanical ventilation. An infusion of 6% albumin was administrated by jugular vein (rate of 5 μL/min) as a fluid support. To access the left ventricle (LV), a 1.0 French pressure-conductance catheter (PVR-1035, Millar Instruments, Houston, TX, USA) was inserted through the right carotid artery. The position of the catheter was carefully adjusted until stable pressure-volume loops were obtained. The data were measured using an MPVS ultra pressure–volume analysis system (Millar Instruments) connected to a PowerLab 4/35 data acquisition system (AD Instruments, Colorado Springs, CO, USA) using LabChart 8 Pro software (AD Instruments). After baseline pressure-volume loops were recorded, occlusion measurements were obtained by inferior vena cava occlusion using a cotton-tipped applicator by direct occlusion using a small forceps after opening the chest. All analyses were performed using LabChart 8.0 Pro (ADInstruments). Measurements were calibrated based on echocardiographic measurements. Hemodynamic parameters and systolic and diastolic indices calculated were: CO, cardiac output; dP/dt−EDV, dP/dtmax−end-diastolic volume relation; dP/dtmax, peak rate of pressure rise; −dP/dtmin, peak rate of pressure decline; Ea, arterial elastance (measure of ventricular afterload); EDP, end-diastolic pressure; EDPVR slope, end-diastolic PV relation slope; EDV, end-diastolic volume; Ees/max, end-systolic elastance (slope of the end-systolic relationship); EF, ejection fraction; efficiency (SW/PV area); ESP, end-systolic pressure; ESV, end-systolic volume; HR, heart rate; MAP, mean arterial pressure; PRSW, SV, stroke volume; SW, stroke work; Tau (W), relaxation time constant calculated by Weiss method (regression of log (pressure) [77]. At the end of the physiologic experiments, animals (under deep anesthesia) were euthanized/perfused for tissue harvest (kidney and plasma). Tissues were saved and snap-frozen with liquid N2 and stored (−80 °C) for Western blot analysis and in formalin 10% for histological analysis.

### 4.6. Immunostaining, Imaging Acquisition and Analyses for SGLT-2 Quantification

To investigate the expression of SGLT-2 in paraffin-embedded renal (or cardiac) sections, immunofluorescence imaging was performed on kidneys from all groups as previously described [78]. Sections from kidneys were incubated for 45 min at 70 degrees, dewaxed with xylene washes and hydrated by graded ethanol washes of 100% (twice for 3 min), 95%, 80% and 70% followed by water immersion (twice for 4 min). Next, slides were steamed for 45 min with 1× citrate Antigen Retrieval Buffer (ab93678). Sections were rinsed with PBS, permeabilized with 0.2% triton x-100 (Sigma-Aldrich, St. Louis, MO, USA) for 30 min blocking with 10% donkey serum (45 min) and incubated overnight with primary antibody Anti-SGLT2 (sc-393350) in blocking solution. On the next day after wash, the slides with PBS were incubated for 1 h with conjugated secondary antibodies: Donkey anti-Mouse IgG, Alexa Fluor 555 (ThermoFisher Scientific A-10037; 1:400). Nuclei were stained with DAPI before mounting with ProLong Gold Antifade (Invitrogen P36934) and coverslips. For quantification, slides were scanned at 20× magnification using the Olympus VS120–L100 Virtual Slide Microscope (Tokyo, Japan) and five images from randomly selected heart sections (10 magnification) were used to quantify SGLT2 using Image J.

### 4.7. Blood Urea Nitrogen (BUN),Creatinine (CRE) and Glucose (Glu) Quantification from the Plasma

Blood from mice fasted overnight was collected with anticoagulant (Heparin) syringes directly from left ventricle of the mice under terminal anesthetic. Plasma BUN, creatinine and glucose were quantitated using a Vitros 5600 analyzer (Ortho, Rochester, NY, USA) in the Pathology Research Resource Veterinary Clinical Pathology Core Laboratory (University of Miami, Miami, FL, USA). BUN is measured via a colorimetric assay and creatinine is measured through a two-point rate enzymatic assay. The analyzer is maintained per manufacturer’s recommendations.

### 4.8. Data Analyses and Statistics

For comparison, mice were separated by sex. Comparisons were made with their respective age-matched WT controls and between groups from different backgrounds. All data were expressed as mean ± SD or SEM. Shapiro–Wilk tests were performed on each parameter to check normality. For all parameters with a normal distribution, we used parametric analysis of variance (two-way ANOVA) followed by Turkey’s post-hoc test or *t*-test when comparing two groups. For other parameters, we used nonparametric Kruskal–Wallis tests followed by Dunn post hoc test or Mann–Whitney when comparing two groups (Graphpad 7.3). Differences were considered statistically significant at *p* < 0.05.

## Figures and Tables

**Figure 1 ijms-23-06674-f001:**
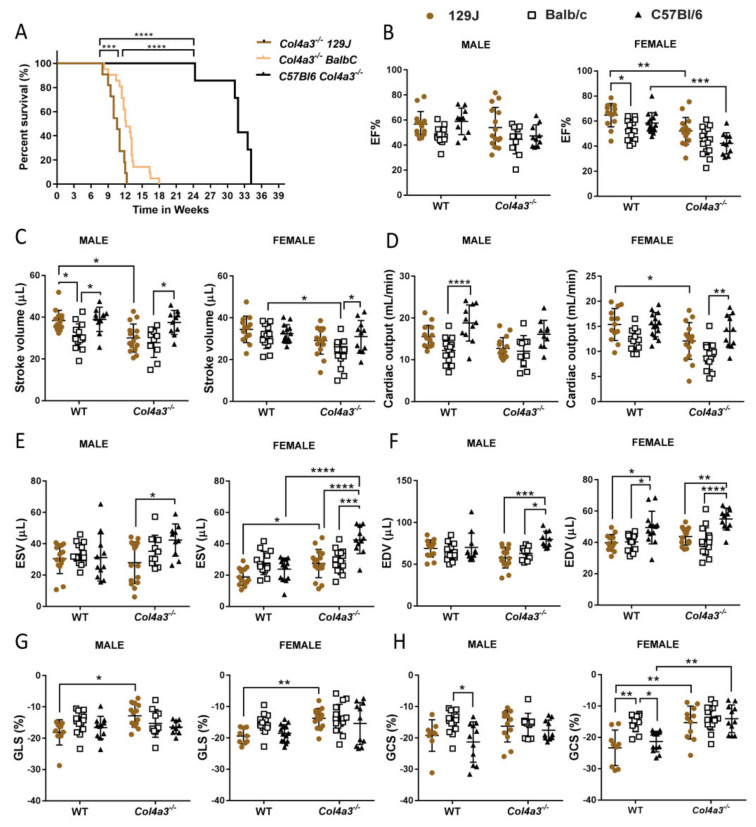
Kaplan–Meier survival curve (**A**) showing significant differences among 129x1/SvJ Alport mice (*n* = 11), C57Bl6 Alport mice (*n* = 7) and Balb/C Alport mice (*n* = 21). Echocardiography of male and female *Col4a3^-/-^* (Alport) and WT mice from 129x1/SvJ (8 weeks old), Balb/C (8 weeks old) and C57Bl/6 (20 weeks old) background showing systolic function parameters: (**B**) EF—ejection fraction (%), (**C**) stroke volume (µL), (**D**) cardiac output (mL/min), (**E**) ESV—end-systolic volume (µL), (**F**) EDV—end-diastolic volume (µL), (**G**) GLS—global longitudinal strain, (**H**) GCS—global circumferential strain. Images were assessed using parasternal long- and short-axis in B-mode. Data are expressed in mean ± SD. * *p* < 0.05, ** *p* < 0.01, *** *p* < 0.001, **** *p* < 0.0001. Groups are: male WT-129x1/SvJ (*n* = 14), Alport 129x1/SvJ (*n* = 15), WT-Balb/C (*n* = 14), Alport-Balb/C (*n* = 10) and WT-C57Bl/6 (*n* = 11) and Alport C57Bl/6 (*n* = 10); female WT-129x1/SvJ (*n* = 13), Alport 129x1/SvJ (*n* = 16), WT-Balb/C (*n* = 13), Alport-Balb/C (*n* = 15) and WT-C57Bl/6 (*n* = 14) and Alport C57Bl/6 (*n* = 11).

**Figure 2 ijms-23-06674-f002:**
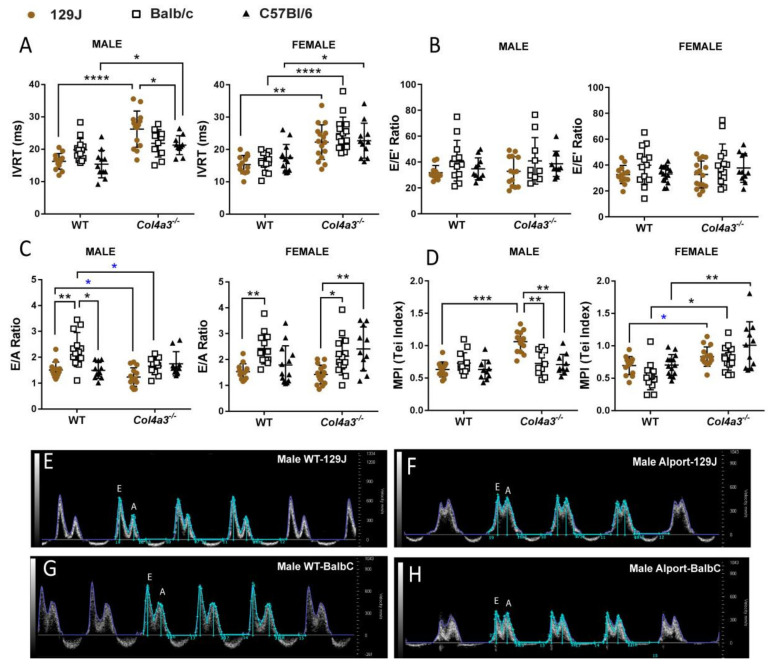
Echocardiography of male and female *Col4a3^-/-^* (Alport) and WT mice from 129x1/SvJ (8 weeks old), Balb/C (8 weeks old) and C57Bl/6 (20 weeks old) background showing diastolic function parameters: (**A**) IVRT- isovolumetric relaxation time (ms), (**B**) E/E′ ratio—ratio between early mitral inflow velocity and mitral annular early diastolic tissue velocity, (**C**) E/A ratio—ratio of early to late mitral inflow velocity, (**D**) MPI—myocardial performance index or Tei index, (**E**) representative images of mitral pulsed wave Doppler flow in male WT-129J mice and (**F**) Alport-129J mice displaying E and A wave measurements. It is possible to observe a more symmetrical relationship between the E and A wave on the Alport-129J mouse indicating a decrease in E/A ratio. (**G**) Representative images of mitral pulsed wave Doppler flow in male WT-Balb/C mice and (**H**) Alport-Balb/C mice displaying E and A wave measurements. Alport-Balb/C mice presented a more symmetrical relationship between E and A wave indicating a decrease in E/A ratio. Images were assessed using four-chamber view. Data are expressed in mean ± SD. * *p* < 0.05, ** *p* < 0.01, *** *p* < 0.001, **** *p* < 0.0001. * *p* < 0.05 (*t*-test or Mann–Whitney) comparing only WT group and its respective Alport group (from the same strain). Groups are: male WT-129x1/SvJ (*n* = 14), Alport 129x1/SvJ (*n* = 15), WT-Balb/C (*n* = 14), Alport-Balb/C (*n* = 10) and WT-C57Bl/6 (*n* = 11) and Alport C57Bl/6 (*n* = 10); female WT-129x1/SvJ (*n* = 13), Alport 129x1/SvJ (*n* = 16), WT-Balb/C (*n* = 13), Alport-Balb/C (*n* = 15) and WT-C57Bl/6 (*n* = 14) and Alport C57Bl/6 (*n* = 11).

**Figure 3 ijms-23-06674-f003:**
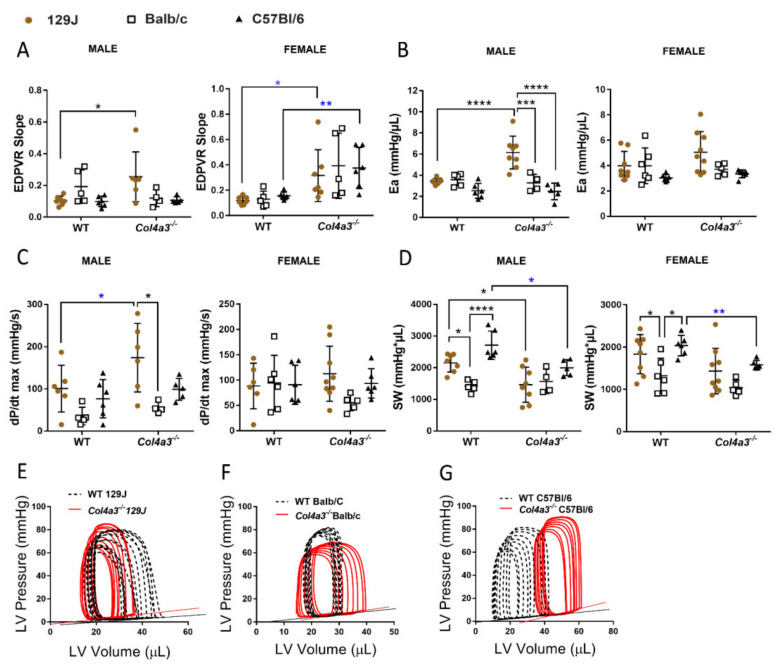
Invasive hemodynamic measurements of male and female *Col4a3^-/-^* (Alport) and WT mice from 129x1/SvJ (8 weeks old), Balb/C (8 weeks old) and C57Bl/6 (20 weeks old) background: (**A**) linear EDVPR—end-diastolic pressure-volume relationship, (**B**) Ea—arterial elastance (**C**) dP/dT_max_ (**D**) SW—stroke work. Representative image of pressure-volume loops representing the changes in pressure and volume of the left ventricle during cardiac cycle comparing females WT (black) and Alport mice (red) from 129x1/SvJ (**E**), Balb/C (**F**) and C57 Bl/6 (**G**) during vena cava occlusion. Data are expressed in mean ± SD. * *p* < 0.05, *** *p* < 0.001, **** *p* < 0.0001. * *p* < 0.05 (*t*-test or Mann–Whitney), ** *p* < 0.01 (*t*-test or Mann–Whitney). Groups are: male WT-129x1/SvJ (*n* = 7), Alport 129x1/SvJ (*n* = 6), WT-Balb/C (*n* = 5), Alport-Balb/C (*n* = 4) and WT-C57Bl/6 (*n* = 6) and Alport C57Bl/6 (*n* = 5); female WT-129x1/SvJ (*n* = 8), Alport 129x1/SvJ (*n* = 9), WT-Balb/C (*n* = 6), Alport-Balb/C (*n* = 6) and WT-C57Bl/6 (*n* = 6) and Alport C57Bl/6 (*n* = 5).

**Figure 4 ijms-23-06674-f004:**
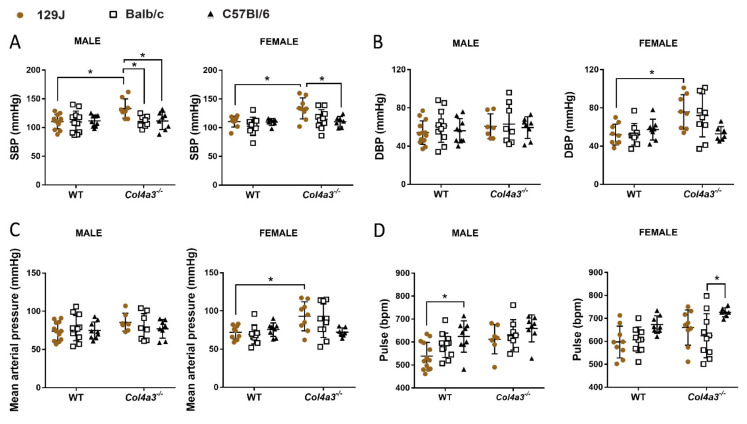
Blood pressure (BP) of male and female *Col4a3^-/-^* (Alport) and WT mice from 129x1/SvJ (8 weeks old), Balb/C (8 weeks old) and C57Bl/6 (20 weeks old) background measure by tail cuff. (**A**) SBP = systolic blood pressure (mmHg), (**B**) DBP = diastolic blood pressure (mmHg), (**C**) mean arterial pressure (mmHg), (**D**) pulse (bpm), bpm = XX. Data are expressed in mean ± SD. * *p* < 0.05. Groups are: male WT-129x1/SvJ (*n* = 12), Alport 129x1/SvJ (*n* = 8), WT-Balb/C (*n* = 11), Alport-Balb/C (*n* = 8) and WT-C57Bl/6 (*n* = 9) and Alport C57Bl/6 (*n* = 8); female WT-129x1/SvJ (*n* = 9), Alport 129x1/SvJ (*n* = 9), WT-Balb/C (*n* = 9), Alport-Balb/C (*n* = 10) and WT-C57Bl/6 (*n* = 8) and Alport C57Bl/6 (*n* = 7).

**Figure 5 ijms-23-06674-f005:**
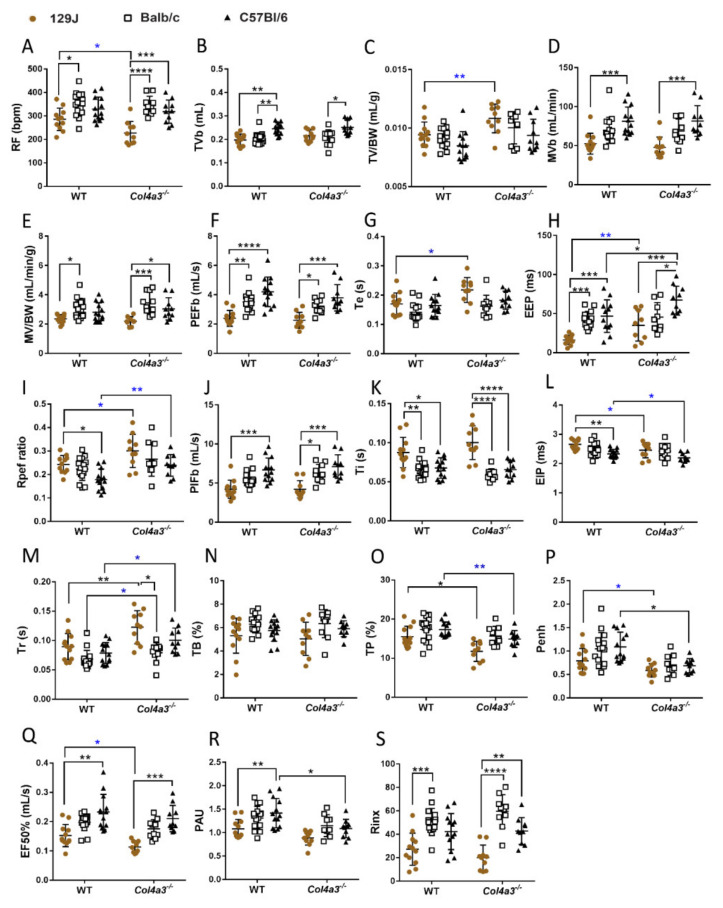
Whole body plethysmography (WBP) of male *Col4a3^-/-^* (Alport) and WT mice from 129x1/SvJ (8 weeks old), Balb/C (8 weeks old) and C57Bl/6 (20 weeks old) background showing parameter from respiratory function: (**A**) RF = respiratory frequency (bpm), (**B**) TVb = tidal volume (mL), (**C**) TVb/BW = tidal volume corrected by body weight (mL/g), (**D**) MVb = minute volume (mL/min), (**E**) MVb/BW = minute volume corrected by body weight (mL/min/g), (**F**) PEFb = time to peak expiratory flow (mL/s), (**G**) Te = expiratory time (s), (**H**) EEP = end expiratory pause (ms), (**I**) Rpef ratio = ratio of time to peak expiratory flow (PEFb) relative to expiratory time (Te), (**J**) PIFb = time to peak inspiratory flow mL/s), (**K**) Ti = inspiratory time (s), (**L**) EIP = end inspiratory pause (ms), (**M**) Tr = relaxation time (s), (**N**) TB = time of brake (%), (**O**) TP = time of pause (%), (**P**) Penh = enhanced pause, (**Q**) EF50% = flow rate which 50% of the tidal volume on an individual breath has been expelled (mL/s), (**R**) PAU, (**S**) Rinx. Data are expressed in mean ± SD. * *p* < 0.05, ** *p* < 0.01, *** *p* < 0.001, **** *p* < 0.0001. * *p* < 0.05, ** *p* < 0.01 (*t*-test or Mann–Whitney). Groups are: WT-129x1/SvJ (*n* = 12), Alport 129x1/SvJ (*n* = 10), WT-Balb/C (*n* = 16), Alport-Balb/C (*n* = 10) and WT-C57Bl/6 (*n* = 13) and Alport C57Bl/6 (*n* = 10).

**Figure 6 ijms-23-06674-f006:**
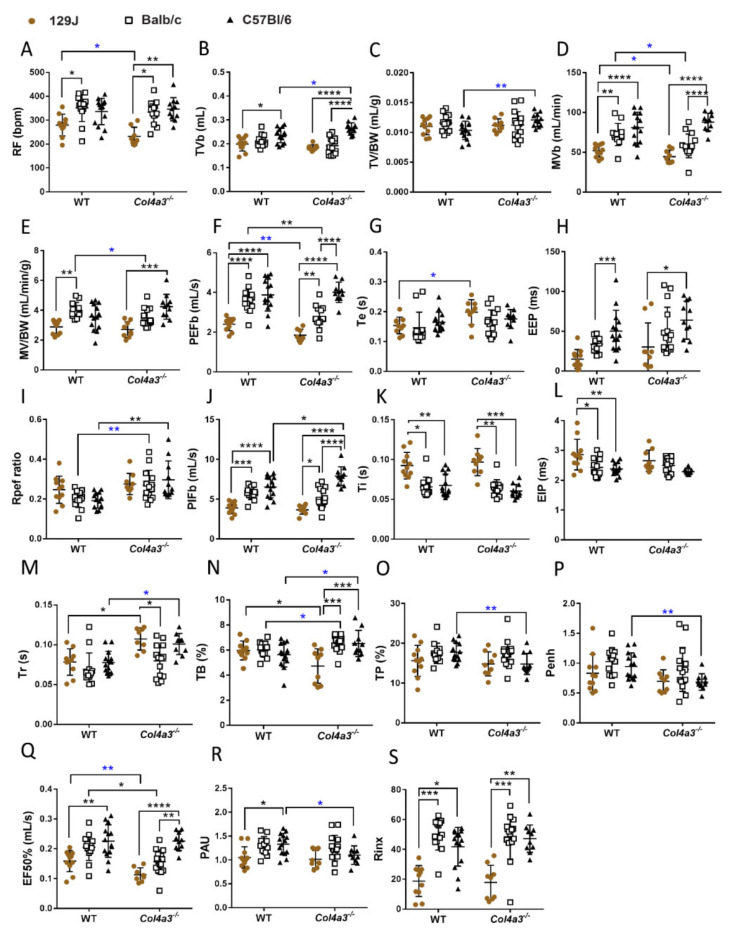
Whole body plethysmography (WBP) of females *Col4a3^-/-^* (Alport) and WT mice from 129x1/SvJ (8 weeks old), Balb/C (8 weeks old) and C57Bl/6 (20 weeks old) background showing parameter from respiratory function: (**A**) RF = respiratory frequency (bpm), (**B**) TVb = tidal volume (mL), (**C**) TVb/BW = tidal volume corrected by body weight (mL/g), (**D**) MVb = minute volume (mL/min), (**E**) MVb/BW = minute volume corrected by body weight (mL/min/g), (**F**) PEFb = time to peak expiratory flow (mL/s), (**G**) Te = expiratory time (s), (**H**) EEP = end expiratory pause (ms), (**I**) Rpef ratio = ratio of time to peak expiratory flow (PEFb) relative to expiratory time (Te), (**J**) PIFb = time to peak inspiratory flow mL/s), (**K**) Ti = inspiratory time (s), (**L**) EIP = end inspiratory pause (ms), (**M**) Tr = relaxation time (s), (**N**) TB = time of brake (%), (**O**) TP = time of pause (%), (**P**) Penh = Enhanced pause, (**Q**) EF50% = flow rate which 50% of the tidal volume on an individual breath has been expelled (mL/s), (**R**) PAU, (**S**) Rinx. Data are expressed in mean ± SD. * *p* < 0.05, ** *p* < 0.01, *** *p* < 0.001, **** *p* < 0.0001. * *p* < 0.05, ** *p* < 0.01 (*t*-test or Mann–Whitney). Groups are: WT-129x1/SvJ (*n* = 11), Alport 129x1/SvJ (*n* = 9), WT-Balb/C (*n* = 13), Alport-Balb/C (*n* = 15) and WT-C57Bl/6 (*n* = 14) and Alport C57Bl/6 (*n* = 11).

**Figure 7 ijms-23-06674-f007:**
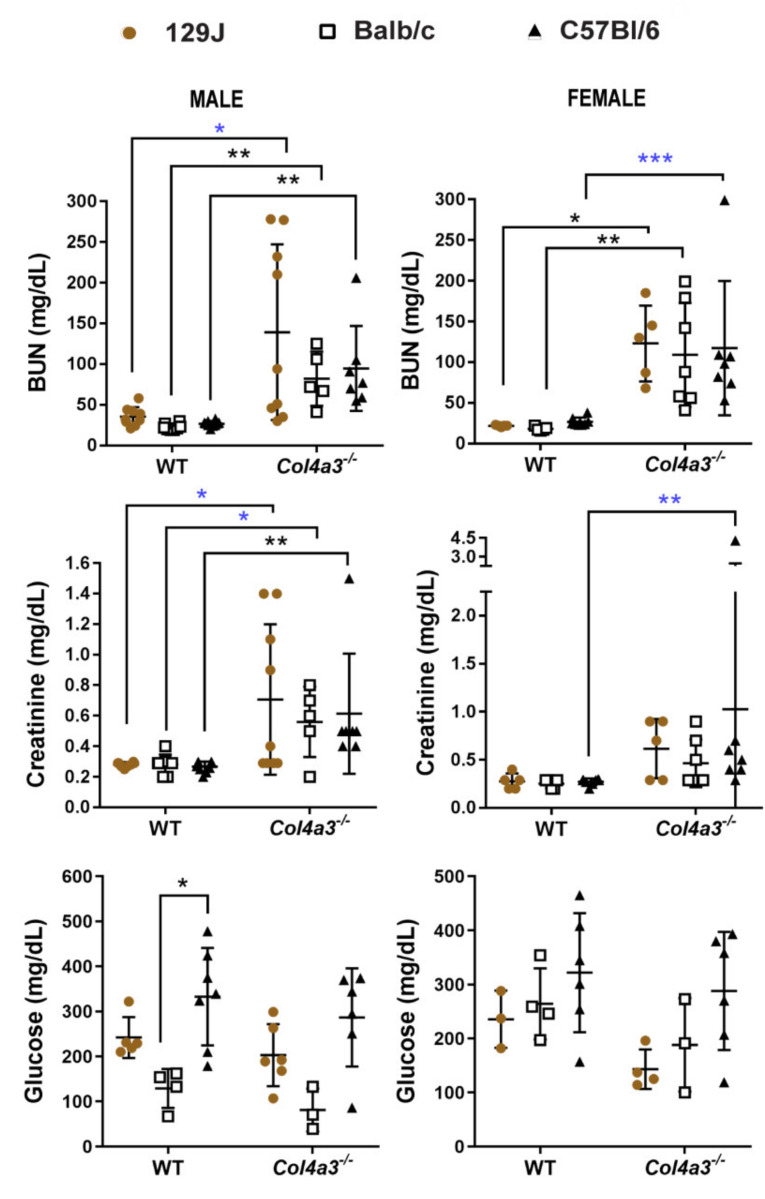
Blood urea (BUN), creatinine (CRE) and glucose (Glu) levels from male and female *Col4a3^-/-^* (Alport) and WT mice from 129x1/SvJ (8 weeks old), Balb/C (8 weeks old) and C57Bl/6 (20 weeks old) background. Data are expressed in mean ± SD. * *p* < 0.05, ** *p* < 0.01. * *p* < 0.05, ** *p* < 0.01, *** *p* = 0.0003 (*t*-test or Mann–Whitney). For BUN and CRE, groups are: males WT-129x1/SvJ (*n* = 9), Alport 129x1/SvJ (*n* = 9), WT-Balb/C (*n* = 8), Alport-Balb/C (*n* = 5) and WT-C57Bl/6 (*n* = 10) and Alport C57Bl/6 (*n* = 7); females WT-129x1/SvJ (*n* = 5), Alport 129x1/SvJ (*n* = 5), WT-Balb/C (*n* = 5), Alport-Balb/C (*n* = 7) and WT-C57Bl/6 (*n* = 8) and Alport C57Bl/6 (*n* = 7). For Glu, groups are males WT-129x1/SvJ (*n* = 5), Alport 129x1/SvJ (*n* = 6), WT-Balb/C (*n* = 4), Alport-Balb/C (*n* = 3) and WT-C57Bl/6 (*n* = 7) and Alport C57Bl/6 (*n* = 6); females WT-129x1/SvJ (*n* = 3), Alport 129x1/SvJ (*n* = 4), WT-Balb/C (*n* = 4), Alport-Balb/C (*n* = 3) and WT-C57Bl/6 (*n* = 6) and Alport:C57Bl/6 (*n* = 6).

**Figure 8 ijms-23-06674-f008:**
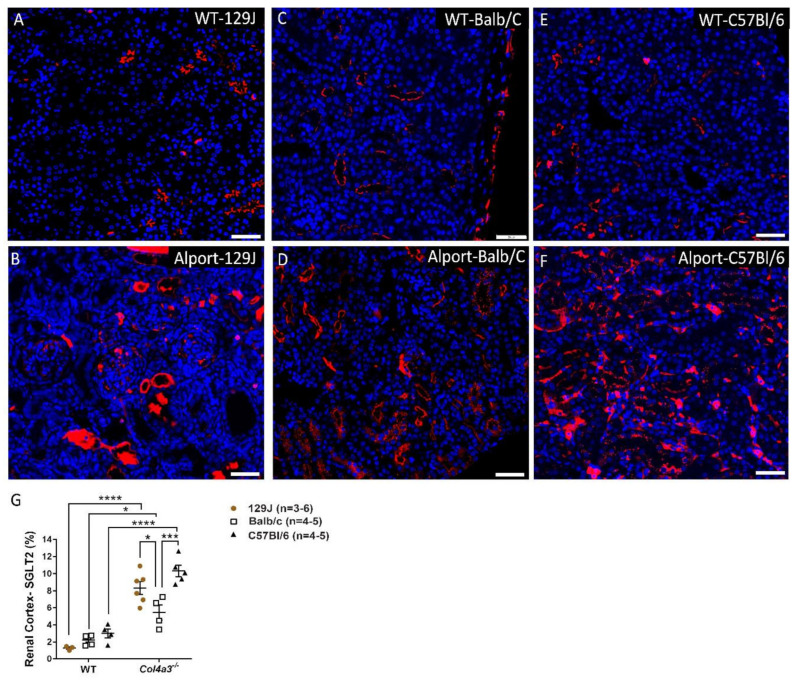
SGLT2 expression in the kidney by immunofluorescence from males *Col4a3^-/-^* (Alport) and WT mice from 129x1/SvJ (8 weeks old), Balb/C (8 weeks old) and C57Bl/6 (20 weeks old) background. (**A**) Representative images of SGLT2 expression in renal cortex from WT-129J mice, (**B**) Alport-129J, (**C**) WT-Balb/C, (**D**) Alport Balb/C, (**E**) WT-C57bl/6, (**F**) Alport C57Bl/6, (**G**) graph showing differences between groups. Data are expressed in mean ± SEM. * *p* < 0.05, *** *p* < 0.001, **** *p* < 0.0001. Scale bar: 50 µm. Groups are: WT-129x1/SvJ (*n* = 3), Alport 129x1/SvJ (*n* = 6), WT-Balb/C (*n* = 5), Alport-Balb/C (*n* = 4) and WT-C57Bl/6 (*n* = 4) and Alport C57Bl/6 (*n* = 5).

**Figure 9 ijms-23-06674-f009:**
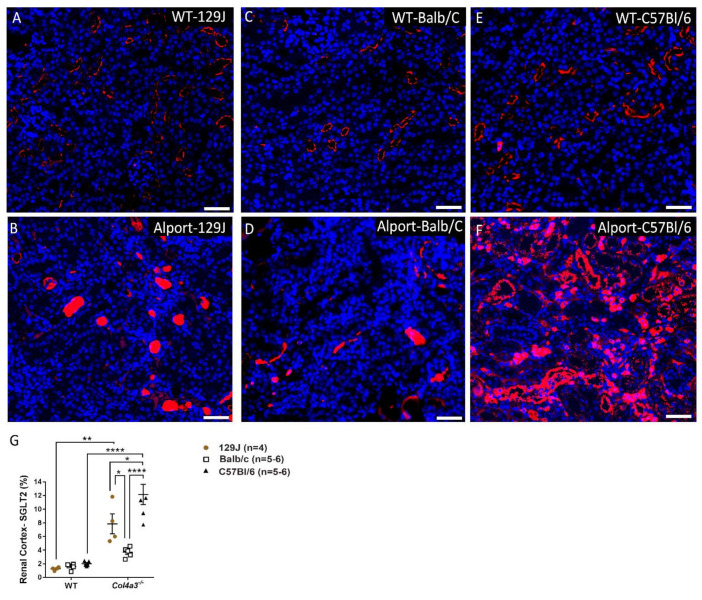
SGLT2 expression in the kidney by immunofluorescence from females *Col4a3^-/-^* (Alport) and WT mice from 129x1/SvJ (8 weeks old), Balb/C (8 weeks old) and C57Bl/6 (20 weeks old) background. (**A**) Representative images of SGLT2 expression in renal cortex from WT-129J mice, (**B**) Alport-129J, (**C**) WT-Balb/C, (**D**) Alport Balb/C, (**E**) WT-C57bl/6, (**F**) Alport C57Bl/6, (**G**) graph showing differences between groups. Data are expressed in mean ± SEM. * *p* < 0.05, ** *p* < 0.01, **** *p* < 0.0001. Scale bar: 50 µm. Groups are: WT-129x1/SvJ (*n* = 4), Alport 129x1/SvJ (*n* = 5), WT-Balb/C (*n* = 6), Alport-Balb/C (*n* = 6) and WT-C57Bl/6 (*n* = 5) and Alport C57Bl/6 (*n* = 6).

**Table 1 ijms-23-06674-t001:** Echocardiography parameters of LV function, pulse wave Doppler, tissue Doppler and morphometric measurements in male mice from 129J, Balb/C and C57Bl/6 background.

Echocardiography Parameters	129J 8 Weeks Old		Balb/C 8 Weeks Old		C57Bl/6 20 Weeks Old	
Systolic Function						
	WT (*n* = 14)	*Col4a3^-/-^*(*n* = 15)	WT (*n* = 14)	*Col4a3^-/-^* (*n* = 10)	WT (*n* = 11)	*Col4a3^-/-^* (*n* = 10)
EF (%)	56.74 ± 10.08	54.04 ± 15.97	48.02 ± 6.63	44.65 ± 11.61	58.97 ± 10.56	47.4 ± 8.8
SV (µL)	38.4 ± 4.97	30.06 ± 6.74 *	30.08 ± 6.05 ^†,€^	27.85 ± 7.06	38.99 ± 5.83	37.5 ± 5.79 ^†^
CO (µL/min)	15.55 ± 2.70	12.74 ± 2.58 ^††††^	12.24 ± 3.06	12.06 ± 3.76	18.82 ± 4.32	16.1 ± 3.34
FS (%)	29.77 ± 7.28	28.38 ± 10.93	23.91 ± 3.99	22.04 ± 6.46	30.12 ± 7.67	23.78 ± 5.41
ESV (µL)	30.47 ± 9.53	27.92 ± 13.31 ^¥^	33.44 ± 6.85	35.23 ± 10.18	31.24 ± 15.11	42.36 ± 10.24
EDV (µL)	68.88 ± 10.21	57.97 ± 12.45 ^¥¥¥^	62.64 ± 8.25	63.08 ± 7.93 ^†^	65.96 ± 9.77	79.86 ± 9.58
HR (bpm)	413.54±40.29 ^¥¥¥^	424.94±40.05	412.78±44.68 ^††^	447.71±42.53	495.34±48.61	446.64 ± 44.42
GLS (%)	−18.12 ± 4.04	−12.85 ± 3.6 *	−14.9 ± 3.55	−15.27 ± 4.4	−16.62 ± 3.6	−16.44 ± 2.12
GCS (%)	−19.22 ± 5.04	−16.26 ± 5.04	−14.89 ± 3.55 ^†^	−15.91 ± 3.80	−21.3 ± 6.45	−17.52 ± 3.22
Diastolic function						
	WT (*n* = 14)	*Col4a3^-/-^*(*n* = 15)	WT (*n* = 14)	*Col4a3^-/-^* (*n* = 10)	WT (*n* = 11)	*Col4a3^-/-^* (*n* = 10)
A′ velocity (mms/s)	−22.05 ± 5.14	−18.38 ± 3.84	−18.43 ± 5.60	−16.9 ± 6.12	−22.59 ± 7.4	−16.47 ± 3.15
E′ velocity (mm/s)	−21.25 ± 5.73	−15.91 ± 4.16	−20.99 ± 8.57	−18.22 ± 5.6	−22.97 ± 5.68	−20.52 ± 4.66
AET (ms)	54.89 ± 6.10	45.32 ± 5.99 *^,€^	55.66 ± 8.26	56.31 ± 11.07	50.68 ± 9.36	50.48 ± 5.49
IVCT (ms)	17.38 ± 4.09	22.29 ± 4.10 ^¥¥^	18.11 ± 6.36	17.64 ± 5.9	13.88 ± 4.85	13.61 ± 5.33
IVRT (ms)	16.25 ± 2.44	26.22 ± 5.59 ****	19.80 ± 3.49	21.83 ± 4.24	15.38 ± 4.23	21.25 ± 2.93 *^,¥^
MV A velocity (mm/s)	441.59 ± 64.77	403.32 ± 88.07	369.08 ± 104.75 ^††^	427.26 ± 145.15	537.07 ± 153.3	467.34 ± 109.2
MV E velocity(mm/s)	655.76 ± 90.86	490.88 ± 162.26 *	796.53 ± 83.88	686.76 ± 186.83 ^€€^	763.44 ± 115.15	762 ± 59 ± 136.31 ^¥¥¥¥^
E/A ratio	1.51 ± 0.30	1.24 ± 0.36	2.3 ± 1.66 ^€€,†^	1.66 ± 0.33	1.49 ± 0.31	1.75 + 0.47
DT (ms)	23.66 ± 3.68	21.14 ± 4.69	26.97 ± 7.6	26.22 ± 6.23	22.07 ± 5.11	29.67 ± 5.84 *^,¥^
MV E/E′ ratio	31.74 ± 5.53	32.77 ± 11.28	41.06 ± 15.73	40.89 ± 18.01	34.71 ± 8.53	38.58 ± 9.79
Mean Velocity (MV VTI, mm/s)	357.44 ± 58.47	302.7 ± 77.60	414.46 ± 55.58	351.97 ± 68.80	428.65 ± 81.06	384.62 ± 75.82
Mean Gradient (MV VTI, mmHg)	0.52 ± 0.16	0.39 ± 0.21	0.70 ± 0.19	0.51 ± 0.20	0.76 ± 0.28	0.61 ± 0.24
Peak Velocity (MV VTI, mm/s)	670.60 ± 83.69	522.73 ± 151.4 ^¥^	807.11 ± 76.82	693.46 ± 187.52	768.38 ± 116.04	767.23 ± 133.64
Peak Gradient (MV VTI, mmHg)	1.83 ± 0.45	1.18 ± 0.71 ^¥^	2.64 ± 0.52	2.05 ± 1.06	2.42 ± 0.72	2.42 ± 0.86
NFT (ms)	89.47 ± 9.24	93.37 ± 13.65	94.22 ± 14.46	96.03 ± 13.92	82.04 ± 13.59	86.31 ± 12.36
A′/E′	1.06 ± 0.14	1.16 ± 0.2 ^¥¥¥^	0.92 ± 0.23	0.96 ± 0.30	1.00 ± 0.3	0.74 ± 0.12
E′/A′	0.96 ± 0.13	0.87 ± 0.18 ^¥¥¥^	1.15 ± 0.30	1.15 ± 0.38	1.08 ± 0.3	1.38 ± 0.22
LV MPI (Tei Index)	0.63 ± 0.12	1.06 ± 0.16 ***	0.73 ± 0.16	0.73 ± 0.19 ^€€^	0.63 ± 0.15	0.71 ± 0.16 ^¥¥^
Cardiac dimensions						
	WT (*n* = 14)	*Col4a3^-/-^*(*n* = 15)	WT (*n* = 14)	*Col4a3^-/-^* (*n* = 10)	WT (*n* = 11)	*Col4a3^-/-^* (*n* = 10)
Area, d (mm^2^)	21.19 ± 1.72	18.97 ± 2.60	20.53 ± 2.71	19.75 ± 2.63	20.43 ± 2.78	21.77 ± 2.25
Area, s (mm^2^)	14.02 ± 1.39	13.59 ± 3.24	14.54 ± 2.20	14.18 ±2.48	12.45 ± 2.59	15.18 ± 6.61
Endo major, d (mm)	7.19 ± 0.36	6.95 ± 0.29	7.27 ± 0.53	6.91 ± 0.48	7.21± 0.40	7.22 ± 0.38
Endo major, s (mm)	6.33 ± 0.32	6.19 ± 0.63	6.64 ± 0.52	6.28 ± 0.50	6.24 ± 0.49	6.62 ± 0.43
Epi major, d (mm)	7.34 ±0.35	7.70 ± 0.46	7.81 ± 0.55	7.43 ± 0.48	7.84 ± 0.32	7.89 ± 0.37
Epi major, s (mm)	6.90 ± 0.35	6.84 ± 0.59	7.16 ± 0.54	6.88 ± 0.46	6.99 ± 0.38	7.32 ± 0.42
LV mass (mg)	106.80±17.0	104.96±15.45	89.44 ± 12.25 ^†††^	99.41 ± 17.17 ^†^	118.38±19.38	123.62±14.13
LV mass/BW (mg/g)	4.82 ± 0.64	5.68 ± 0.79 *	4.1 ± 0.47	5.01 ± 1.01	4.11 ± 0.61	4.57 ± 0.66 ^¥¥^
LVAW, d (mm)	0.77 ± 0.08	0.86 ± 0.18	0.66 ± 0.13	0.83 ± 0.19	0.82 ± 0.13	0.74 ± 0.12
LVAW, s (mm)	1.03 ± 0.15	1.16 ± 0.28	0.94 ± 0.21^†^	1.08 ± 0.29	1.21 ± 0.23	1.05 ± 0.19
LVPW, d (mm)	0.78 ± 0.17	0.79 ± 0.14 ^€^	0.66 ± 0.09	0.64 ± 0.101	0.72 ± 0.08	0.72 ± 0.06
LVPW, s (mm)	0.97 ± 0.16	0.98 ± 0.15	0.85 ±0.08^††^	0.84 ± 0.09	1.08 ±0.18	0.97 ± 0.12
LVID, d (mm)	3.84 ± 0.23	3.56 ± 0.35 ^¥¥¥^	3.88 ± 0.34	3.81 ± 0.19 ^†^	4.0 ± 0.39	4.2 ± 0.17
LVID, s (mm)	2.88 ± 0.43	2.68 ± 0.53 ^¥^	2.95 ± 0.26	2.94 ± 0.37	2.82 ± 0.62	3.21 ±0.29

EF: ejection fraction, SV: stroke volume, CO: cardiac output, FS: fractional shortening, EDV: end-diastolic volume, ESV: end-systolic volume, HR: heart rate, GLS: global longitudinal strain, GCS: global circumferential strain, AET: aortic ejection time, IVCT: isovolumetric contraction time; IVRT: isovolumetric relaxation time, MV: mitral valve, DT: deceleration time, VTI: velocity time integral, NFT: no flow time; LV MPI: left ventricular myocardial performance index (Tei index), d, diastolic; s, systolic, LV: left ventricle, LVAW: left ventricular anterior wall, LVPW, left ventricular posterior wall, LVID, left ventricular internal diameter. Values are mean ± SD. * *p* < 0.05, *** *p* < 0.001 and **** *p* < 0.0001 compared *Col4a3^-/-^* with their respective control (same strain), ^†^ *p* < 0.05, ^††^ *p* < 0.01, ^†††^ *p* < 0.001, ^††††^ *p* < 0.0001 differences between Balb/C strain and C57Bl/6 strain (WT vs. WT and *Col4a3^-/-^* vs. *Col4a3^-/-^*), ^€^ *p* < 0.05, ^€€^ *p* < 0.01, differences between Balb/C strain and 129J (WT vs. WT and *Col4a3^-/-^* vs. *Col4a3^-/-^*), ^¥^ *p* < 0.05, ^¥¥^ *p* < 0.01, ^¥¥¥^ *p* < 0.001, ^¥¥¥¥^ *p* < 0.0001 difference between 129J and C57Bl/6 (WT vs. WT and *Col4a3^-/-^* vs. *Col4a3^-/-^*).

**Table 2 ijms-23-06674-t002:** Echocardiography parameters of LV function, pulse wave Doppler, tissue Doppler and morphometric measurements in female mice from 129J, Balb/C and C57Bl/6 background.

Echocardiography Parameters	129J 8 Weeks Old		Balb/C 8 Weeks Old		C57Bl/6 20 Weeks Old	
Systolic function						
	WT (*n* = 13)	*Col4a3^-/-^* (*n* = 16)	WT (*n* = 13)	*Col4a3^-/-^* (*n* = 15)	WT (*n* = 14)	*Col4a3^-/-^* (*n* = 11)
EF (%)	64.75 ± 9.3	52.17 ± 10.73 **	53.1± 8.35 ^€^	44.09 ± 10.97	58.48 ± 8.31	42.21 ± 8.49 ***
SV (µL)	34.49 ± 6.32	29.15 ± 6.53	30.94 ± 5.6	22.93 ± 6.46 *	32.46 ± 4.27	31.02 ± 7.72 ^†^
CO (µL/min)	15.38 ± 3.17	12.1 ± 3.65 *	12.37 ± 2.085	9.06 ± 2.68	15.34 ± 2.42	14.03 ± 3.3 ^††^
FS (%)	35.07 ± 6.65	26.56 ± 7.01 *	27.07 ± 5.28	21.55 ± 6.22	30.63 ± 6.17	20.64 ± 4.9 **
ESV (µL)	18.87 ± 5.59 ^€^	27.42 ± 9.4 *^,¥¥¥¥^	27.72 ± 7.44	29.07 ± 7.22 ^†††^	23.85 ± 7.12	42.53 ± 8.82 ****
EDV (µL)	53.35 ± 6.9	56.57 ± 10.58	58.66 ± 8.88	52 ± 8.68 ^††^	57.7 ± 8.12	71.86 ± 11.23
HR (bpm)	448.88 ± 43.06	418.16 ± 38.21	429.53 ± 23.30 ^†^	412.80 ± 48.60	477.75 ± 40.40	450.77 ± 39.50
GLS (%)	−19.39 ± 2.45	−13.73 ± 3.16 **	−15.38 ± 3.16	−13.72 ± 4.48	−18.51 ± 2.78	−15.39 ± 6.68
GCS (%)	−23.34 ± 5.68	−15.27 ± 5.26 **	−15.3 ± 2.87 ^€€,†^	−13.84 ± 3.82	−21.32 ± 3.28	−14.06 ± 4.42 **
Diastolic function						
	WT (*n* = 13)	*Col4a3^-/-^*(*n* = 16)	WT (*n* = 13)	*Col4a3^-/-^* (*n* = 15)	WT (*n* = 14)	*Col4a3^-/-^*(*n* = 11)
A′ velocity (mms/s)	−21.48 ± 3.54	−21.23 ± 6.21	−17.17 ± 5.12	−19.05 ± 4.58	−20.67 ± 7.13	−18.27 ± 4.32
E′ velocity (mm/s)	−23.25 ± 4.67	−19.76 ± 6.74	−20.22 ± 6.50	−18.11 ± 7.14	−24.87 ± 4.40	−21.62 ± 4.76
AET (ms)	48.78 ± 5.80	49.50 ± 5.57	60.83 ± 8.08 ^€€€€^	56.42 ± 7.15 ^€^	50.42 ± 4.26 ^†††^	47.73 ± 6.8 ^†^
IVCT (ms)	17.18 ± 4.0	19.82 ± 3.80	15.36 ± 6.43	22.12 ± 7.94 *	15.73 ± 5.05	22.40 ± 7.25
IVRT (ms)	15.32 ± 2.83	22.27 ± 5.35 **	16.01 ± 2.88	24.92 ± 5.10 ****	17.44 ± 4.10	22.73 ± 5.40 *
MV A velocity (mm/s)	463.80 ± 80.98	434.08 ± 110.99	330.59 ± 68.99 ^†^	353.43 ± 130.98	474.39 ±159.92	359.06 ± 163.32
MV E velocity(mm/s)	739.94 ± 141.21	589.35 ± 117.47 ^¥¥^	774.37 ± 141.74	685.81 ± 145.70	816.64 ± 189.27	783.16 ± 160.59
E/A ratio	1.53 ± 0.30 ^€€^	1.42 ± 0.36 ^€,¥¥^	2.43 ± 0.60	2.13 ± 0.72	1.79 ± 0.74	2.41 ± 0.84
DT (ms)	18.59 ± 2.94 ^€€€^	20.16 ± 3.58	27.23 ± 4.48	25.43 ± 7.49	19.44 ± 5.11 ^††^	18.32 ± 4.99 ^††^
MV E/E′ ratio	32.63 ± 7.04	32.73 ± 10.50	39.83 ± 14.56	40.47 ± 15.85	33.13 ± 6.52	37.71 ± 11.1
Mean Velocity (MV VTI, mm/s)	428.37 ± 79.15	348.63 ± 68.52	376.39 ± 89.42	365.36 ± 82.51	444.68 ± 78.08	404.03 ± 95.04
Mean Gradient (MV VTI, mmHg)	0.76 ± 0.3	0.50 ± 0.20	0.63 ± 0.27	0.56 ± 0.27	0.81 ± 0.32	0.69 ± 0.32
Peak Velocity (MV VTI, mm/s)	748.39 ± 140.78	605.08 ± 112.36 ^¥^	777.75 ± 146.00	682.06 ± 145.09	815.08 ± 188	775.7 ± 173.64
Peak Gradient (MV VTI, mmHg)	2.32 ± 0.89	1.51 ± 0.59 ^¥^	2.51 ± 1.94	1.94 ± 0.86	2.79 ± 1.39	2.52 ± 1.02
NFT (ms)	82.05 ± 6.42	90.88 ± 11.23	91.97 ± 9.01	103.86 ± 14.90	85.62 ± 9.02	94.28 ± 11.3
A′/E′	0.95 ± 0.23	1.14 ± 0.29	0.88 ± 0.2	1.17± 0.39	0.84 ± 0.29	0.85 ± 0.132
E′/A′	1.10 ± 0.25	0.94 ± 0.28	1.19 ± 0.27	0.95 ± 0.33	1.33 ± 0.47	1.20 ± 0.21
LV MPI (Tei Index)	0.69 ± 0.14	0.83 ± 0.15	0.53 ± 0.21	0.81 ± 0.18 *	0.70 ± 0.16	1.0 ± 0.36 **
Cardiac dimensions						
	WT (*n* = 13)	*Col4a3^-/-^* (*n* = 16)	WT (*n* = 13)	*Col4a3^-/-^* (*n* = 15)	WT (*n* = 14)	*Col4a3^-/-^* (*n* = 11)
Area, d (mm2)	17.31 ± 1.39	18.16 ± 1.35	18.02 ± 1.37	18.04 ± 2.27	19.68 ± 3.6	21.80 ± 1.50 ^¥¥¥,††^
Area, s (mm2)	10.36 ± 1.34	13.04 ± 10.65 *	11.90 ± 1.60	12.71 ± 2.44	12.14 ± 3.09	15.88 ± 2.0 ***^,¥,†††^
Endo major, d (mm)	6.41 ± 0.42 ^¥¥^	6.53 ± 0.33 ^¥¥^	6.83 ± 0.33	6.88 ± 0.44	7.02 ± 0.75	7.33 ± 0.37
Endo major, s (mm)	5.43 ± 0.22 ^€€,¥¥^	5.72 ± 0.43 ^€,¥¥^	6.16 ± 0.38	6.33 ± 0.40	6.13 ± 0.73	6.67 ± 0.44
Epi major, d (mm)	6.89 ± 0.38 ¥¥	7.13 ± 0.33 ¥¥	7.35 ± 0.31	7.39 ± 0.46	7.57 ± 0.82	7.89 ± 0.31
Epi major, s (mm)	5.96 ± 0.25 ^¥¥¥,€^	6.40 ± 0.44 ¥¥¥	6.71 ± 0.37	6.88 ± 0.44	6.94 ± 0.50	7.27 ± 0.32
LV mass (mg)	79.35 ± 12.15	83.70 ± 13.55 ¥	86.64 ± 16.41	74.33 ± 20.60 ^†^	84.81 ± 14.19	107.03 ± 15.5
LV mass/BW (mg/g)	4.5 ± 0.45	5.11 ± 0.87	4.88 ± 0.83	4.33 ± 1.03	4.21 ± 0.61	4.90 ± 0.65
LVAW, d (mm)	0.73 ± 0.088	0.72 ± 0.054	0.69 ± 0.132	0.66 ± 0.113	0.69 ± 0.13	0.66 ± 0.11
LVAW, s (mm)	0.95 ± 0.15	0.94 ± 0.12	0.95 ± 0.14	0.88 ± 0.13	1.10 ± 0.12	0.96 ± 0.22
LVPW, d (mm)	0.66 ± 0.09	0.71 ± 0.12	0.67 ± 0.10	0.66 ± 0.15	0.73 ± 0.12	0.74 ±0.11
LVPW, s (mm)	0.94 ± 0.11	0.89 ± 0.11	0.90 ± 0.12	0.81 ± 0.21	1.04 ± 0.15	0.96 ± 0.17
LVID, d (mm)	3.42 ± 0.19	3.51 ± 0.31 ^¥¥¥¥^	3.65 ± 0.27	3.48 ± 0.25 ^††††^	3.63 ± 0.27	4.03 ± 0.27 **
LVID, s (mm)	2.42 ± 0.27	2.78 ± 0.38 ^¥^	2.78 ± 0.28	2.77 ± 0.32 ^†^	2.57 ± 0.38	3.2 ± 0.31 ***

EF: ejection fraction, SV: stroke volume, CO: cardiac output, FS: fractional shortening, EDV: end-diastolic volume, ESV: end-systolic volume, HR: heart rate, GLS: global longitudinal strain, GCS: global circumferential strain, AET: aortic ejection time, IVCT: isovolumetric contraction time; IVRT: isovolumetric relaxation time, MV: mitral valve, DT: deceleration time, VTI: velocity time integral, NFT: no flow time; LV MPI: left ventricular myocardial performance index (Tei index), d, diastolic; s, systolic, LV: left ventricle, LVAW: left ventricular anterior wall, LVPW, left ventricular posterior wall, LVID, left ventricular internal diameter. Values are mean ± SD. * *p* < 0.05, ** *p* < 0.01, *** *p* < 0.001 and **** *p* < 0.0001 compared *Col4a3^-/-^* with their respective control (same strain), ^†^ *p* < 0.05, ^††^ *p* < 0.01, ^†††^ *p* < 0.001, ^††††^ *p* < 0.0001 differences between Balb/C strain and C57Bl/6 strain (WT vs. WT and *Col4a3^-/-^* vs. *Col4a3^-/-^*), **^€^** *p* < 0.05, **^€€^** *p* < 0.01, **^€€€^** *p* < 0.001, **^€€€€^** *p* < 0.0001 differences between Balb/C strain and 129J (WT vs. WT and *Col4a3^-/-^* vs. *Col4a3^-/-^*), ^¥^ *p* < 0.05, ^¥¥^ *p* < 0.01, ^¥¥¥^ *p* < 0.001, ^¥¥¥¥^ *p* < 0.0001 difference between 129J and C57Bl/6 (WT vs. WT and *Col4a3^-/-^* vs. *Col4a3^-/-^*).

## Data Availability

Not applicable.

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
