# Peer review of "Col4a3-/- Mice on Balb/C Background Have Less Severe Cardiorespiratory Phenotype and SGLT2 Over-Expression Compared to 129x1/SvJ and C57Bl/6 Backgrounds"

_ijms, 2022, doi:10.3390/ijms23126674_

Round 1
Reviewer 1 Report
No further comments.
Author Response
Thank you for endorsing our manuscript with no further comments.
Reviewer 2 Report
I previously reviewed this manuscript:
"The authors have carried out an extensive characterization of cardiac and respiratory function in several mouse models of Alport syndrome. Their data collection is comprehensive. They have assessed SGLT2 expression in the renal cortex of the several murine models. The methods employed are exhaustively characterized and implemented. What remains tenuous is an attempt to relate their findings to Alport’s syndrome in humans. The chief features of Alport’s syndrome include renal disease with hematuria, hearing loss, and ocular findings. Lung and renal disease are not noted features. The authors provide convincing evidence that murine models with alteration in the collagen IV gene show cardiac and respiratory abnormalities. They have nicely characterized those abnormalities. Their SGLT2 expression data is solely in renal cortex. It would be interesting for them to assess SGLT2 expression in myocardial tissue.
What would be of most benefit to the reader would be an exposition of the phenotypic characteristics of the mouse strains employed, with characterization of degree of abnormality in expression of the collagen IV gene, clearly both in males and females, since the genotypic alteration is X linked. It is reasonable to point out the similarities with human Alport’s syndrome and the definitely the differences. There is nothing wrong with exploring abnormalities in these murine models, but let us not try too hard to connect to the human syndrome. Suffice it to say that these genomic abnormalities give rise to abnormalities in cardiac and respiratory parameters, perhaps relating to stiffness of connective tissue?
If SGLT2 is to be the primary focus of this work, then do studies with SGLT2 inhibitors and SGLT2 knockout mice and present this data."
The authors have really not responded to my suggestions. That is all right, but they really should distinguish very clearly between Alports syndrome in humans and the murine model. Perhaps a table showing the cardiac, respiratory and renal impairments comparing humans to mice.
Author Response
We respectfully disagree that we did not respond to the reviewer’s suggestions. While we did not add a table to summarize the different and common features between mouse and human Alport Syndrome (as we believe a general information summary table is not ideal in a research article), we extensively revised the manuscript to include these good points brought up by the reviewer. In the submitted manuscript we highlight in the introduction, results and discussion all this information showing that we were responsive. We believe that this presentation of the information is more effective that a small summary table in our research manuscript that is already busy with 9 data Figures, 2 data Tables, and 4 Supplemental data Tables.
Round 2
Reviewer 2 Report
This is a rereview. The manuscript is much improved. The only suggestion I have is to document the human reports of aortic disease in Alports syndrome.
Author Response
Thank you for your feedback. We now add to the Introduction:
"There have been several reports of aortic abnormalities in young males with AS, including aortic dilation, thoracic and abdominal aortic aneurysms and aortic dissection [20-23]. There is a potential molecular link between collagen IV α5 variants and aortic disease, since the collagen IV α5α5α6 network is normally expressed in the basement membranes of the aortic musculature in non-Alport subjects [24,25] and in wild type but not Alport mice [20]. If aortic disease is truly a complication of AS, very little is known about its incidence, prevalence or natural history (reviewed in [26]). "
This manuscript is a resubmission of an earlier submission. The following is a list of the peer review reports and author responses from that submission.
Round 1
Reviewer 1 Report
The authors analyzed the strain-dependent cardiorespiratory phenotypes and their relation with renal SGLT-2 expression in the mouse model of Alport syndrome. Overall, the manuscript is written well, however, needs the following questions to be addressed.
1) What are the factors or causes that lead to differential abilities in getting Alport syndrome. At least perform a few experiments that demonstrate such differentiation among mouse strains.
2) SGLT-2 inhibitors are the known drug to treat hyperglycemia or diabetes by excretion of glucose into the urine. A high level of SGLT-2 in diabetic conditions increases the glucotoxicity in the tubules, endothelium and, several orgens. SGLT-2 inhibitors inhibit glucotoxicity hence providing protection against glucose toxicity.
Here, my question is what is the pathogenic role of SGLT-2 in AS. Besides glucotoxicity, what does SGLT-2 do in AS? Perform key experiments showing key mechanisms. Does it have glucose independent mechanism? Also, analyze the level of glucose in your experiment.
3) Does SGLT-2 overexpression or inhibition cause hemodynamic alteration to changes in blood glucose level?
4) Figure 8 and Figure 8; Use related lecithin to SGTL-2 expression in different kidney compartments.
5) Introduction: Include it
Empagliflozin is shown protective against diabetic kidney disease by inhibiting EMT
Reviewer 2 Report
The authors have carried out an extensive characterization of cardiac and respiratory function in several mouse models of Alport syndrome. Their data collection is comprehensive. They have assessed SGLT2 expression in the renal cortex of the several murine models. The methods employed are exhaustively characterized and implemented. What remains tenuous is an attempt to relate their findings to Alport’s syndrome in humans. The chief features of Alport’s syndrome include renal disease with hematuria, hearing loss, and ocular findings. Lung and renal disease are not noted features. The authors provide convincing evidence that murine models with alteration in the collagen IV gene show cardiac and respiratory abnormalities. They have nicely characterized those abnormalities. Their SGLT2 expression data is solely in renal cortex. It would be interesting for them to assess SGLT2 expression in myocardial tissue.
What would be of most benefit to the reader would be an exposition of the phenotypic characteristics of the mouse strains employed, with characterization of degree of abnormality in expression of the collagen IV gene, clearly both in males and females, since the genotypic alteration is X linked. It is reasonable to point out the similarities with human Alport’s syndrome and the definitely the differences. Perhaps they can do this in a Table.
There is nothing wrong with exploring abnormalities in these murine models, but let us not try too hard to connect to the human syndrome. Suffice it to say that these genomic abnormalities give rise to abnormalities in cardiac and respiratory parameters, perhaps relating to stiffness of connective tissue?
If SGLT2 is to be the primary focus of this work, then do studies with SGLT2 inhibitors and SGLT2 knockout mice and present this data.